# ROS: A GNN-BASED RELAX-OPTIMIZE-AND-SAMPLE FRAMEWORK FOR MAX-$k$-CUT PROBLEMS

## ABSTRACT

The Max-$k$-Cut problem is a fundamental combinatorial optimization challenge that generalizes the classic $\mathcal{NP}$-complete Max-Cut problem. While relaxation techniques are commonly employed to tackle Max-$k$-Cut, they often lack guarantees of equivalence between the solutions of the original problem and its relaxation. To address this issue, we introduce the Relax-Optimize-and-Sample (ROS) framework. In particular, we begin by relaxing the discrete constraints to the continuous probability simplex form. Next, we pre-train and fine-tune a graph neural network model to efficiently optimize the relaxed problem. Subsequently, we propose a sampling-based construction algorithm to map the continuous solution back to a high-quality Max-$k$-Cut solution. By integrating geometric landscape analysis with statistical theory, we establish the consistency of function values between the continuous solution and its mapped counterpart. Extensive experimental results on random regular graphs and the Gset benchmark demonstrate that the proposed ROS framework effectively scales to large instances with up to $20,000$ nodes in just a few seconds, outperforming state-of-the-art algorithms. Furthermore, ROS exhibits strong generalization capabilities across both in-distribution and out-of-distribution instances, underscoring its effectiveness for large-scale optimization tasks.

## 1 INTRODUCTION

The *Max-$k$-Cut problem* involves partitioning the vertices of a graph into $k$ disjoint subsets in such a way that the total weight of edges between vertices in different subsets is maximized. This problem represents a significant challenge in combinatorial optimization and finds applications across various fields, including telecommunication networks (Eisenblätter, 2002; Gui et al., 2018), data clustering (Poland & Zeugmann, 2006; Ly et al., 2023), and theoretical physics (Cook et al., 2019; Coja-Oghlan et al., 2022). The Max-$k$-Cut problem is known to be $\mathcal{NP}$-complete, as it generalizes the well-known *Max-Cut problem*, which is one of the 21 classic $\mathcal{NP}$-complete problems identified by Karp (2010).

Significant efforts have been made to develop methods for solving Max-$k$-Cut problems (Nath & Kuhnle, 2024). Ghaddar et al. (2011) introduced an exact branch-and-cut algorithm based on semidefinite programming, capable of handling graphs with up to 100 vertices. For larger instances, various polynomial-time approximation algorithms have been proposed. Goemans & Williamson (1995) addressed the Max-Cut problem by first solving a semi-definite relaxation to obtain a fractional solution, then applying a randomization technique to convert it into a feasible solution, resulting in a 0.878-approximation algorithm. Building on this, Frieze & Jerrum (1997) extended the approach to Max-$k$-Cut, offering feasible solutions with approximation guarantees. de Klerk et al. (2004) further improved these guarantees, while Shinde et al. (2021) optimized memory usage. Despite their strong theoretical performance, these approximation algorithms involve solving computationally intensive semi-definite programs, rendering them impractical for large-scale Max-$k$-Cut problems. A variety of heuristic methods have been developed to tackle the scalability challenge. For the Max-Cut problem, Burer et al. (2002) proposed rank-two relaxation-based heuristics, and Goudet et al. (2024) introduced a meta-heuristic approach using evolutionary algorithms. For Max-$k$-Cut, heuristics such as genetic algorithms (Li & Wang, 2016), greedy search (Gui et al., 2018), multiple operator heuristics (Ma & Hao, 2017), and local search (Garvardt et al., 2023) have been proposed. While these

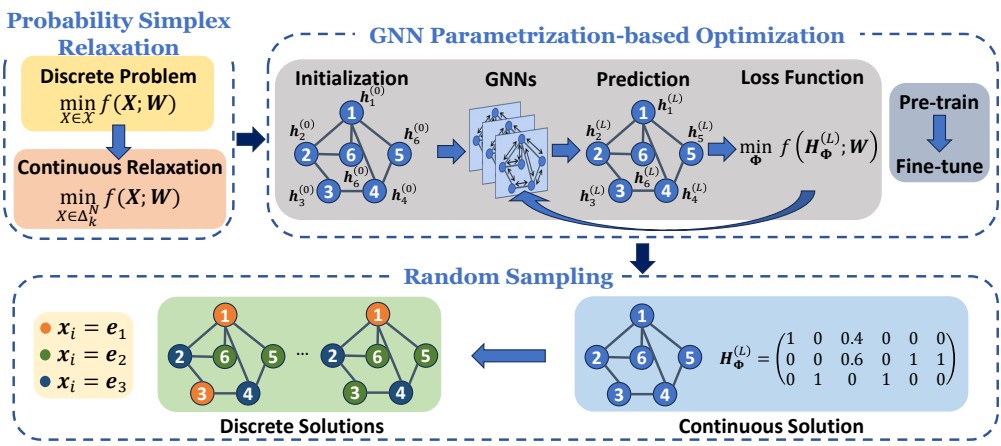

Figure 1: The Relax-Optimize-and-Sample framework.

heuristics can handle much larger Max-$k$-Cut instances, they often struggle to balance efficiency and solution quality.

Recently, *machine learning* techniques have gained attention for enhancing optimization algorithms (Bengio et al., 2021; Gasse et al., 2022; Chen et al., 2024). Several studies, including Khalil et al. (2017); Barrett et al. (2020); Chen et al. (2020); Barrett et al. (2022), framed the Max-Cut problem as a sequential decision-making process, using reinforcement learning to train policy networks for generating feasible solutions. However, RL-based methods often suffer from extensive sampling efforts and increased complexity in action space when extended to Max-$k$-Cut, and hence entails significantly longer training and testing time. Karalias & Loukas (2020) focuses on subset selection, including Max-Cut as a special case. It trains a *graph neural network* (GNN) to produce a distribution over subsets of nodes of an input graph by minimizing a probabilistic penalty loss function. After the network has been trained, a randomized algorithm is employed to sequentially decode a valid Max-Cut solution from the learned distribution. A notable advancement by Schuetz et al. (2022) reformulated Max-Cut as a quadratic unconstrained binary optimization (QUBO), removing binarity constraints to create a differentiable loss function. This loss function was used to train a GNN, followed by a simple projection onto integer variables after unsupervised training. The key feature of this approach is solving the Max-Cut problem during the training phase, eliminating the need for a separate testing stage. Although this method can produce high-quality solutions for Max-Cut instances with millions of nodes, the computational time remains significant due to the need to optimize a parameterized GNN from scratch. The work of Tönshoff et al. (2022) first formulated the Max-Cut problem as a *constraint satisfaction problem* and then proposed a novel GNN-based reinforcement learning approach. This method outperforms prior neural combinatorial optimization techniques and conventional search heuristics. However, to the best of our knowledge, it is limited to unweighted Max-$k$-Cut problems.

In this work, we propose a GNN-based *Relax-Optimize-and-Sample* (ROS) framework for efficiently solving the Max-$k$-Cut problem with arbitrary edge weights. The framework is depicted in Figure 1. Initially, the Max-$k$-Cut problem is formulated as a discrete optimization task. To handle this, we introduce *probability simplex relaxations*, transforming the discrete problem into a continuous one. We then optimize the relaxed formulation by training parameterized GNNs in an unsupervised manner. To further improve efficiency, we apply *transfer learning*, utilizing pre-trained GNNs to warm-start the training process. Finally, we refine the continuous solution using a *random sampling algorithm*, resulting in high-quality Max-$k$-Cut solutions.

The key contributions of our work are summarized as follows:

- **Novel Framework.** We propose a scalable ROS framework tailored to the weighted Max-$k$-Cut problem with arbitrary signs, built on solving continuous relaxations using efficient learning-based techniques.

- **Theoretical Foundations.** We conduct a rigorous theoretical analysis of both the relaxation and sampling steps. By integrating geometric landscape analysis with statistical theory, we demonstrate the consistency of function values between the continuous solution and its sampled discrete counterpart.
- **Superior Performance.** Comprehensive experiments on public benchmark datasets show that our framework produces high-quality solutions for Max-$k$-Cut instances with up to $20,000$ nodes in just a few seconds. Our approach significantly outperforms state-of-the-art algorithms, while also demonstrating strong generalization across various instance types.

## 2 PRELIMINARIES

### 2.1 MAX-$k$-CUT PROBLEMS

Let $\mathcal{G} = (\mathcal{V}, \mathcal{E})$ represent an undirected graph with vertex set $\mathcal{V}$ and edge set $\mathcal{E}$. Each edge $(i, j) \in \mathcal{E}$ is assigned an arbitrary weight $\boldsymbol{W}_{ij} \in \mathbb{R}$, which can have any sign. A *cut* in $\mathcal{G}$ refers to a partition of its vertex set. The Max-$k$-Cut problem involves finding a $k$-partition $(\mathcal{V}_1, \ldots, \mathcal{V}_k)$ of the vertex set $\mathcal{V}$ such that the sum of the weights of the edges between different partitions is maximized.

To represent this partitioning, we employ a $k$-dimensional one-hot encoding scheme. Specifically, we define a $k \times N$ matrix $\boldsymbol{X} \in \mathbb{R}^{k \times N}$ where each column represents a one-hot vector. The Max-$k$-Cut problem can be formulated as:

$$\max_{\boldsymbol{X} \in \mathbb{R}^{k \times N}} \quad \frac{1}{2} \sum_{i=1}^{N} \sum_{j=1}^{N} \boldsymbol{W}_{ij} \left(1 - \boldsymbol{X}_{\cdot i}^{\top} \boldsymbol{X}_{\cdot j}\right) \tag{1}$$
$$\text{s. t.} \quad \boldsymbol{X}_{\cdot j} \in \{\boldsymbol{e}_1, \boldsymbol{e}_2, \ldots, \boldsymbol{e}_k\} \quad \forall j \in \mathcal{V},$$

where $\boldsymbol{X}_{\cdot j}$ denotes the $j^{th}$ column of $\boldsymbol{X}$, $\boldsymbol{W}$ is a symmetric matrix with zero diagonal entries, and $\boldsymbol{e}_\ell \in \mathbb{R}^k$ is a one-hot vector with the $\ell^{th}$ entry set to $1$. This formulation aims to maximize the total weight of edges between different partitions, ensuring that each node is assigned to exactly one partition, represented by the one-hot encoded vectors. We remark that weighted Max-$k$-Cut problems with arbitrary signs is a generalization of classic Max-Cut problems and arise in many interesting applications (De Simone et al., 1995; Poland & Zeugmann, 2006; Hojny et al., 2021).

### 2.2 GRAPH NEURAL NETWORKS

GNNs are powerful tools for learning representations from graph-structured data. GNNs operate by iteratively aggregating information from a node's neighbors, enabling each node to capture increasingly larger sub-graph structures as more layers are stacked. This process allows GNNs to learn complex patterns and relationships between nodes, based on their local connectivity.

At the initial layer ($l = 0$), each node $i \in \mathcal{V}$ is assigned a feature vector $\boldsymbol{h}_i^{(0)}$, which typically originates from node features or labels. The representation of node $i$ is then recursively updated at each subsequent layer through a parametric aggregation function $f_{\boldsymbol{\Phi}^{(l)}}$, defined as:

$$\boldsymbol{h}_i^{(l)} = f_{\boldsymbol{\Phi}^{(l)}} \left(\boldsymbol{h}_i^{(l-1)}, \{\boldsymbol{h}_j^{(l-1)} : j \in \mathcal{N}(i)\}\right), \tag{2}$$

where $\boldsymbol{\Phi}^{(l)}$ represents the trainable parameters at layer $l$, $\mathcal{N}(i)$ denotes the set of neighbors of node $i$, and $\boldsymbol{h}_i^{(l)}$ is the node's embedding at layer $l$ for $l \in \{1, 2, \cdots, L\}$. This iterative process enables the GNN to propagate information throughout the graph, capturing both local and global structural properties.

## 3 A RELAX-OPTIMIZE-AND-SAMPLE FRAMEWORK

In this work, we leverage continuous optimization techniques to tackle Max-$k$-Cut problems, introducing a novel ROS framework. Acknowledging the inherent challenges of discrete optimization, we begin by relaxing the problem to probability simplices and concentrate on optimizing this relaxed version. To achieve this, we propose a machine learning-based approach. Specifically, we model the

relaxed problem using GNNs, pre-training the GNN on a curated graph dataset before fine-tuning it on the specific target instance. After obtaining high-quality solutions to the relaxed continuous problem, we employ a random sampling procedure to derive a discrete solution that preserves the same objective value.

## 3.1 PROBABILITY SIMPLEX RELAXATIONS

To simplify the formulation of the problem (1), we remove constant terms and negate the objective function, yielding an equivalent formulation expressed as follows:

$$\min_{\boldsymbol{X} \in \mathcal{X}} \quad f(\boldsymbol{X}; \boldsymbol{W}) \coloneqq \operatorname{Tr}(\boldsymbol{X} \boldsymbol{W} \boldsymbol{X}^{\top}), \qquad (\mathbf{P})$$

where $\mathcal{X} \coloneqq \left\{ \boldsymbol{X} \in \mathbb{R}^{k \times N} : \boldsymbol{X}_{\cdot j} \in \{\boldsymbol{e}_1, \boldsymbol{e}_2, \dots, \boldsymbol{e}_k\}, \forall j \in \mathcal{V} \right\}$. It is important to note that the matrix $\boldsymbol{W}$ is indefinite due to its diagonal entries being set to zero.

Given the challenges associated with solving the discrete problem $\mathbf{P}$, we adopt a naive relaxation approach, obtaining the convex hull of $\mathcal{X}$ as the Cartesian product of $N$ $k$-dimensional probability simplices, denoted by $\Delta_k^N$. Consequently, the discrete problem $\mathbf{P}$ is relaxed into the following continuous optimization form:

$$\min_{\boldsymbol{X} \in \Delta_k^N} \quad f(\boldsymbol{X}; \boldsymbol{W}). \qquad (\overline{\mathbf{P}})$$

Before optimizing problem $\overline{\mathbf{P}}$, we will characterize its *geometric landscape*. To facilitate this, we introduce the following definition.

**Definition 1.** *Let $\overline{\boldsymbol{X}}$ denote a point in $\Delta_k^N$. We define the neighborhood induced by $\overline{\boldsymbol{X}}$ as follows:*

$$\mathcal{N}(\overline{\boldsymbol{X}}) \coloneqq \left\{ \boldsymbol{X} \in \Delta_k^N \; \middle| \; \sum_{i \in \mathcal{K}(\overline{\boldsymbol{X}}_{\cdot j})} \boldsymbol{X}_{ij} = 1, \quad \forall j \in \mathcal{V} \right\},$$

*where $\mathcal{K}(\overline{\boldsymbol{X}}_{\cdot j}) \coloneqq \{i \in \{1, \dots, k\} \mid \overline{\boldsymbol{X}}_{ij} > 0\}$.*

The set $\mathcal{N}(\overline{\boldsymbol{X}})$ represents a neighborhood around $\overline{\boldsymbol{X}}$, where each point in $\mathcal{N}(\overline{\boldsymbol{X}})$ can be derived by allowing each non-zero entry of the matrix $\overline{\boldsymbol{X}}$ to vary freely, while the other entries are set to zero. Utilizing this definition, we can establish the following theorem.

**Theorem 1.** *Let $\overline{\boldsymbol{X}}$ denote a globally optimal solution to $\overline{\mathbf{P}}$, and let $\mathcal{N}(\overline{\boldsymbol{X}})$ be its induced neighborhood. Then*

$$f(\boldsymbol{X}; \boldsymbol{W}) = f(\overline{\boldsymbol{X}}; \boldsymbol{W}), \quad \forall \boldsymbol{X} \in \mathcal{N}(\overline{\boldsymbol{X}}).$$

Theorem 1 states that for a globally optimal solution $\overline{\boldsymbol{X}}$, every point within its neighborhood $\mathcal{N}(\overline{\boldsymbol{X}})$ shares the same objective value as $\overline{\boldsymbol{X}}$, thus forming a *basin* in the geometric landscape of $f(\boldsymbol{X}; \boldsymbol{W})$. If $\overline{\boldsymbol{X}} \in \mathcal{X}$ (i.e., an integer solution), then $\mathcal{N}(\overline{\boldsymbol{X}})$ reduces to the singleton set $\{\overline{\boldsymbol{X}}\}$. Conversely, if $\overline{\boldsymbol{X}} \notin \mathcal{X}$, there exist $\prod_{j \in \mathcal{V}} |\mathcal{K}(\overline{\boldsymbol{X}}_{\cdot j})|$ unique integer solutions within $\mathcal{N}(\overline{\boldsymbol{X}})$ that maintain the same objective value as $\overline{\boldsymbol{X}}$. This indicates that once a globally optimal solution to the relaxed problem $\overline{\mathbf{P}}$ is identified, it becomes straightforward to construct an optimal solution for the original problem $\mathbf{P}$ that preserves the same objective value.

According to Carlson & Nemhauser (1966), among all globally optimal solutions to the relaxed problem $\overline{\mathbf{P}}$, there is always at least one integer solution. Theorem 1 extends this result, indicating that if the globally optimal solution is fractional, we can provide a straightforward and efficient method to derive its integer counterpart. We remark that it is highly non-trivial to guarantee that the feasible Max-$k$-Cut solution obtained from the relaxation one has the same quality.

**Example**. Consider a Max-Cut problem ($k = 2$) associated with the weight matrix $\boldsymbol{W}$. We optimize its relaxation and obtain the optimal solution $\boldsymbol{X}^{\star}$.

$$\boldsymbol{W} \coloneqq \begin{pmatrix} 0 & 1 & 1 \\ 1 & 0 & 1 \\ 1 & 1 & 0 \end{pmatrix}, \boldsymbol{X}^{\star} \coloneqq \begin{pmatrix} p & 1 & 0 \\ 1-p & 0 & 1 \end{pmatrix},$$

where $p \in [0, 1]$. From the neighborhood $\mathcal{N}(\overline{\boldsymbol{X}})$, We can identify the following integer solutions that maintain the same objective value.

$$\boldsymbol{X}_1^\star = \begin{pmatrix} 0 & 1 & 0 \\ 1 & 0 & 1 \end{pmatrix}, \boldsymbol{X}_2^\star = \begin{pmatrix} 1 & 1 & 0 \\ 0 & 0 & 1 \end{pmatrix}.$$

Given that $\overline{\mathbf{P}}$ is a non-convex program, identifying its global minimum is challenging. Consequently, the following two critical questions arise.

**Q1**. Since solving $\overline{\mathbf{P}}$ to global optimality is $\mathcal{NP}$-hard, how to efficiently optimize $\overline{\mathbf{P}}$ for high-quality solutions?

**Q2**. Given $\overline{\boldsymbol{X}} \in \Delta_k^N \setminus \mathcal{X}$ as a high-quality solution to $\overline{\mathbf{P}}$, can we construct a feasible solution $\hat{\boldsymbol{X}} \in \mathcal{X}$ to $\mathbf{P}$ such that $f(\hat{\boldsymbol{X}}; \boldsymbol{W}) = f(\overline{\boldsymbol{X}}; \boldsymbol{W})$?

We provide a positive answer to **Q2** in Section 3.2, while our approach to addressing **Q1** is deferred to Section 3.3.

## 3.2 RANDOM SAMPLING

Let $\overline{\boldsymbol{X}} \in \Delta_k^N \setminus \mathcal{X}$ be a feasible solution to the relaxation $\overline{\mathbf{P}}$. Our goal is to construct a feasible solution $\boldsymbol{X} \in \mathcal{X}$ for the original problem $\mathbf{P}$, ensuring that the corresponding objective values are equal. Inspired by Theorem 1, we propose a *random sampling* procedure, outlined in Algorithm 1. In this approach, we sample each column $\boldsymbol{X}_{\cdot i}$ of the matrix $\boldsymbol{X}$ from a categorical distribution characterized by the event probabilities $\overline{\boldsymbol{X}}_{\cdot i}$ (denoted as $\text{Cat}(\boldsymbol{x}; \boldsymbol{p} = \overline{\boldsymbol{X}}_{\cdot i})$ in Step 3 of Algorithm 1). This randomized approach yields a feasible solution $\hat{\boldsymbol{X}}$ for $\mathbf{P}$. However, since Algorithm 1 incorporates randomness in generating $\hat{\boldsymbol{X}}$ from $\overline{\boldsymbol{X}}$, the value of $f(\hat{\boldsymbol{X}}; \boldsymbol{W})$ becomes random as well. This raises the critical question: is this value greater or lesser than $f(\overline{\boldsymbol{X}}; \boldsymbol{W})$? We address this question in Theorem 2.

---

**Algorithm 1** Random Sampling

---

1: **Input:** $\overline{\boldsymbol{X}} \in \Delta_k^N$                                     ▷ any feasible solution to $\overline{\boldsymbol{P}}$
2: **for** $i = 1$ to $N$ **do**                                  ▷ each dimension is independent
3:      $\hat{\boldsymbol{X}}_{\cdot i} \sim \text{Cat}(\boldsymbol{x}; \boldsymbol{p} = \overline{\boldsymbol{X}}_{\cdot i})$            ▷ sampling from a categorical distribution
4: **end for**
5: **Output:** $\hat{\boldsymbol{X}} \in \mathcal{X}$                                     ▷ a feasible solution to $\boldsymbol{P}$

---

**Theorem 2.** *Let $\overline{\boldsymbol{X}}$ and $\hat{\boldsymbol{X}}$ denote the input and output of Algorithm 1, respectively. Then, we have* $\mathbb{E}_{\hat{\boldsymbol{X}}}[f(\hat{\boldsymbol{X}}; \boldsymbol{W})] = f(\overline{\boldsymbol{X}}; \boldsymbol{W})$.

Theorem 2 states that $f(\hat{\boldsymbol{X}}; \boldsymbol{W})$ is equal to $f(\overline{\boldsymbol{X}}; \boldsymbol{W})$ in expectation. This implies that the random sampling procedure operates on a fractional solution, yielding Max-$k$-Cut feasible solutions with the same objective values in a probabilistic sense. While the Lovász-extension-based method (Bach et al., 2013) also offers a framework for continuous relaxation, achieving similar theoretical results for arbitrary $k$ and edge weights $W_{i,j} \in \mathbb{R}$ is not always guaranteed. In practice, we execute Algorithm 1 $T$ times and select the solution with the lowest objective value as our best result. We remark that the theoretical interpretation in Theorem 2 distinguishes our sampling algorithm from the existing ones in the literature (Toenshoff et al., 2021; Karalias & Loukas, 2020).

## 3.3 GNN PARAMETRIZATION-BASED OPTIMIZATION

To solve the problem $\overline{\mathbf{P}}$, we propose an efficient learning-to-optimize (L2O) method based on GNN parametrization. This approach reduces the laborious iterations typically required by classical optimization methods (e.g., mirror descent). Additionally, we introduce a "pre-train + fine-tune" strategy, where the model is endowed with prior graph knowledge during the pre-training phase, significantly decreasing the computational time required to optimize $\overline{\mathbf{P}}$.

**GNN Parametrization.** The Max-$k$-Cut problem can be framed as a node classification task, allowing us to leverage GNNs to aggregate node features, and obtain high-quality solutions. Initially, we assign a random embedding $\boldsymbol{h}_i^{(0)}$ to each node $i$ in the graph $\mathcal{G}$, as defined in Section 2. We adopt the GNN architecture proposed by Morris et al. (2019), utilizing an $L$-layer GNN with updates at layer $l$ defined as follows:

$$\boldsymbol{h}_i^{(l)} \coloneqq \sigma\left(\boldsymbol{\Phi}_1^{(l)}\boldsymbol{h}_i^{(l-1)} + \boldsymbol{\Phi}_2^{(l)}\sum_{j\in\mathcal{N}(i)}w_{ji}\boldsymbol{h}_j^{(l-1)}\right),$$

where $\sigma(\cdot)$ is an activation function, and $\boldsymbol{\Phi}_1^{(l)}$ and $\boldsymbol{\Phi}_2^{(l)}$ are the trainable parameters at layer $l$ for $l \in \{1, \ldots, L\}$. This formulation facilitates efficient learning of node representations by leveraging both node features and the underlying graph structure. After processing through $L$ layers of GNN, we obtain the final output $\boldsymbol{H}_{\boldsymbol{\Phi}}^{(L)} \coloneqq [\boldsymbol{h}_1^{(L)}, \ldots, \boldsymbol{h}_N^{(L)}] \in \mathbb{R}^{k\times N}$. A softmax activation function is then applied in the last layer to ensure $\boldsymbol{H}_{\boldsymbol{\Phi}}^{(L)} \in \Delta_k^N$, making the final output feasible for $\overline{\boldsymbol{P}}$.

**"Pre-train + Fine-tune" Optimization.** We propose a "pre-train + fine-tune" framework for learning the trainable weights of GNNs. Initially, the model is trained on a collection of pre-collected datasets to produce a pre-trained model. Subsequently, we fine-tune this pre-trained model for each specific problem instance. This approach equips the model with prior knowledge of graph structures during the pre-training phase, significantly reducing the overall solving time. Furthermore, it allows for out-of-distribution generalization due to the fine-tuning step.

The trainable parameters $\boldsymbol{\Phi} \coloneqq (\boldsymbol{\Phi}_1^{(1)}, \boldsymbol{\Phi}_2^{(1)}, \ldots, \boldsymbol{\Phi}_1^{(L)}, \boldsymbol{\Phi}_2^{(L)})$ in the pre-training phase are optimized using the Adam optimizer with *random initialization*, targeting the objective

$$\min_{\boldsymbol{\Phi}} \quad \mathcal{L}_{\text{pre-training}}(\boldsymbol{\Phi}) \coloneqq \frac{1}{M}\sum_{m=1}^{M} f(\boldsymbol{H}_{\boldsymbol{\Phi}}^{(L)}; \boldsymbol{W}_{\text{train}}^{(m)}),$$

where $\mathcal{D} \coloneqq \{\boldsymbol{W}_{\text{train}}^{(1)}, \ldots, \boldsymbol{W}_{\text{train}}^{(M)}\}$ represents the pre-training dataset. In the fine-tuning phase, for a problem instance represented by $\boldsymbol{W}_{\text{test}}$, the Adam optimizer seeks to solve

$$\min_{\boldsymbol{\Phi}} \quad \mathcal{L}_{\text{fine-tuning}}(\boldsymbol{\Phi}) \coloneqq f(\boldsymbol{H}_{\boldsymbol{\Phi}}^{(L)}; \boldsymbol{W}_{\text{test}}),$$

initialized with the pre-trained parameters.

Moreover, to enable the GNN model to fully adapt to specific problem instances, the pre-training phase can be omitted, enabling the model to be directly trained and tested on the same instance. While this direct approach may necessitate more computational time, it often results in improved performance regarding the objective function. Consequently, users can choose to include a pre-training phase based on the specific requirements of their application scenarios.

## 4 EXPERIMENTS

### 4.1 EXPERIMENTAL SETTINGS

We compare the performance of ROS against traditional methods and L2O algorithms for solving the Max-$k$-Cut problem. Additionally, we assess the impact of the "Pre-train" stage in the GNN parametrization-based optimization. The source code is available at https://anonymous.4open.science/r/ROS_anonymous-1C88/.

**Baseline Algorithms.** We denote our proposed algorithms by ROS and compare them against both traditional algorithms and learning-based methods. When the pre-training step is skipped, we refer to our algorithm as ROS-vanilla. The following traditional Max-$k$-Cut algorithms are considered as baselines: (i) GW (Goemans & Williamson, 1995): an method with a $0.878$-approximation guarantee based on semi-definite relaxation; (ii) BQP (Gui et al., 2018): a local search method designed for binary quadratic programs; (iii) Genetic (Li & Wang, 2016): a genetic algorithm specifically for Max-$k$-Cut problems; (iv) MD: a mirror descent algorithm that addresses the relaxed problem $\overline{\mathbf{P}}$ with a convergence tolerance at $10^{-8}$ and adopts the same random sampling procedure; (v) LPI (Goudet

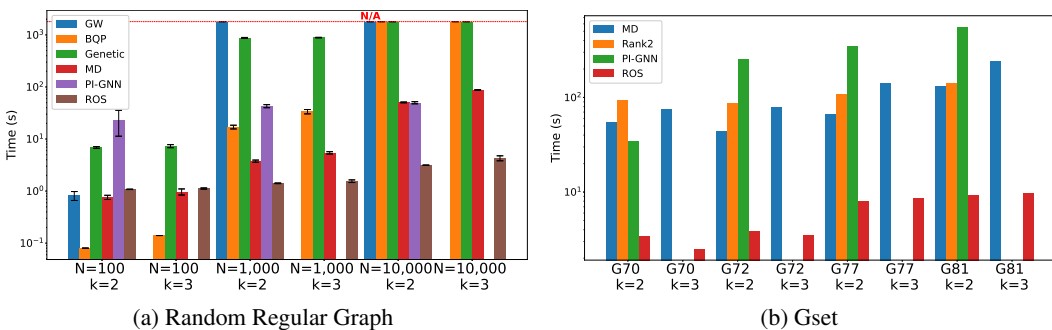


(a) Random Regular Graph          (b) Gset


Figure 2: The computational time comparison of Max-$k$-Cut problems.

et al., 2024): an evolutionary algorithm featuring a large population organized across different islands; (vi) `MOH` (Ma & Hao, 2017): a heuristic algorithm based on multiple operator heuristics, employing various distinct search operators within the search phase. (vii) `Rank2` (Burer et al., 2002): a heuristic based on rank-2 relaxation. For the L2O method, we primarily examine the state-of-the-art baseline: (viii) `PI-GNN` (Schuetz et al., 2022): A cutting-edge L2O method capable of solving QUBO problems in dozens of seconds, delivering commendable performance. It is the first method to eliminate the dependence on large, labeled training datasets typically required by supervised learning approaches.

**Datasets.** The datasets utilized in this paper comprise random regular graphs from Schuetz et al. (2022) and the Gset benchmark from Ye (2003). For the random regular graphs, we employ the `random_regular_graph` from the NetworkX library (Hagberg et al., 2008) to generate $r$-regular graphs, which are undirected graphs in which all nodes have a degree of $r$, with all edge weights equal to 1. The Gset benchmark is constructed using a machine-independent graph generator, encompassing toroidal, planar, and randomly weighted graphs with vertex counts ranging from 800 to 20,000 and edge densities between 2% and 6%. The edge weights in these graphs are constrained to values of 1, 0, or $-1$. Specifically, the training dataset includes 500 3-regular graphs and 500 5-regular graphs, each containing 100 nodes, tailored for the cases where $k = 2$ and $k = 3$, respectively. The testing set for random regular graphs consists 20 3-regular graphs and 20 5-regular graphs for both $k = 2$ and $k = 3$ tasks, with node counts of 100, 1,000, and 10,000, respectively. Moreover, the testing set of Gset encompasses all instances included in the Gset benchmark.

**Model Settings.** `ROS` is designed as a two-layer GNN, with both the input and hidden dimensions set to 100. To address the issue of gradient vanishing, we apply a graph normalization technique as proposed by Cai et al. (2021). The `ROS` model undergoes pre-training using the Adam optimizer with a learning rate of $10^{-2}$ for one epoch. During the fine-tuning stage, the model is further optimized using the same Adam optimizer and learning rate of $10^{-2}$. An early stopping strategy is employed, with a tolerance of $10^{-2}$ and a patience of 100 iterations, terminating training if no improvement is observed over this duration. Finally, in the random sampling stage, we execute Algorithm 1 for $T = 100$ independent trials and return the best solution obtained.

**Evaluation Configuration.** All our experiments were conducted on an NVIDIA RTX 3090 GPU, using Python 3.8.19 and PyTorch 2.2.0.

## 4.2 PERFORMANCE COMPARISON AGAINST BASELINES

### 4.2.1 COMPUTATIONAL TIME

We evaluated the performance of `ROS` against baseline algorithms `GW`, `BQP`, `Genetic`, `MD`, and `PI-GNN` on random regular graphs, focusing on computational time for both the Max-Cut and Max-3-Cut tasks. The experiments were conducted across three problem sizes: $N = 100$, $N = 1,000$, and $N = 10,000$, as illustrated in Figure 2a. Additionally, Figure 2b compares the scalable methods `MD`, `Rank2`, and `PI-GNN` on problem instances from the Gset benchmark with $N \geq 10,000$. "**N/A**" denotes a failure to return a solution within 30 minutes. A comprehensive summary of the results for

Table 1: Objective value comparison of Max-$k$-Cut problems on random regular graphs.

| Methods | N=100 | | N=1,000 | | N=10,000 | |
|---|---|---|---|---|---|---|
| | $k=2$ | $k=3$ | $k=2$ | $k=3$ | $k=2$ | $k=3$ |
| GW | $130.20_{\pm 2.79}$ | – | **N/A** | – | **N/A** | – |
| BQP | $131.55_{\pm 2.42}$ | $239.70_{\pm 1.82}$ | $1324.45_{\pm 6.34}$ | $2419.15_{\pm 6.78}$ | **N/A** | **N/A** |
| Genetic | $127.55_{\pm 2.82}$ | $235.50_{\pm 3.15}$ | $1136.65_{\pm 10.37}$ | $2130.30_{\pm 8.49}$ | **N/A** | **N/A** |
| MD | $127.20_{\pm 2.16}$ | $235.50_{\pm 3.29}$ | $1250.35_{\pm 11.21}$ | $2344.85_{\pm 9.86}$ | $12428.85_{\pm 26.13}$ | $23341.20_{\pm 32.87}$ |
| PI-GNN | $122.75_{\pm 4.36}$ | – | $1263.95_{\pm 21.59}$ | – | $12655.05_{\pm 94.25}$ | – |
| ROS | $128.20_{\pm 2.82}$ | $240.30_{\pm 2.59}$ | $1283.75_{\pm 6.89}$ | $2405.75_{\pm 5.72}$ | $12856.85_{\pm 26.50}$ | $24085.95_{\pm 21.88}$ |

Table 2: Objective value comparison of Max-$k$-Cut problems on Gset instances.

| Methods | G70 (N=10,000) | | G72 (N=10,000) | | G77 (N=14,000) | | G81 (N=20,000) | |
|---|---|---|---|---|---|---|---|---|
| | $k=2$ | $k=3$ | $k=2$ | $k=3$ | $k=2$ | $k=3$ | $k=2$ | $k=3$ |
| MD | 8551 | 9728 | 5638 | 6612 | 7934 | 9294 | 11226 | 13098 |
| Rank2 | 9529 | – | 6820 | – | 9670 | – | 13662 | – |
| PI-GNN | 8956 | – | 4544 | – | 6406 | – | 8970 | – |
| ROS | 8916 | 9971 | 6102 | 7297 | 8740 | 10329 | 12332 | 14464 |

all Gset instances on Max-Cut and Max-3-Cut, including comparisons with state-of-the-art methods LPI and MOH, is presented in Table 3 and Table 4 in the Appendix.

The results depicted in Figure 2a indicate that ROS efficiently solves all problem instances within seconds, even for large problem sizes of $N = 10,000$. In terms of baseline performance, the approximation algorithm GW performs efficiently on instances with $N = 100$, but it struggles with larger sizes such as $N = 1,000$ and $N = 10,000$ due to the substantial computational burden associated with solving the underlying semi-definite programming problem. Heuristic methods such as BQP and Genetic can manage cases up to $N = 1,000$ in a few hundred seconds, yet they fail to solve larger instances with $N = 10,000$ because of the high computational cost of each iteration. Notably, MD is the only method capable of solving large instances within a reasonable time frame; however, when $N$ reaches $10,000$, the computational time for MD approaches 15 times that of ROS. Regarding learning-based methods, PI-GNN necessitates retraining and prediction for each test instance, with test times exceeding dozens of seconds even for $N = 100$. In contrast, ROS solves these large instances in merely a few seconds. Throughout the experiments, ROS consistently completes its tasks in under 10 seconds, requiring only 10% of the computational time utilized by PI-GNN. Figure 2b illustrates the results for the Gset benchmark, where ROS efficiently solves the largest instances in just a few seconds, while other methods, such as Rank2, take tens to hundreds of seconds for equivalent tasks. Remarkably, ROS utilizes only about 1% of the computational time required by PI-GNN.

### 4.2.2 OBJECTIVE VALUE

We also evaluate the performance of ROS on random regular graphs and the Gset benchmark concerning the objective values of Problem (1). The results for the random regular graphs and Gset are presented in Tables 1 and 2, respectively. Note that "–" indicates that the method is unable to handle Max-$k$-Cut problems.

The results for random regular graphs, presented in Table 1, indicate that ROS effectively addresses both $k = 2$ and $k = 3$ cases, producing high-quality solutions even for large-scale problem instances. In contrast, traditional methods such as GW and the L2O method PI-GNN are restricted to $k = 2$ and fail to generalize to the general $k$, i.e., $k = 3$. While GW achieves high-quality solutions for the Max-Cut problem with an instance size of $N = 100$, it cannot generalize to arbitrary $k$ without integrating additional randomized algorithms to yield discrete solutions. Similarly, the L2O method PI-GNN cannot manage $k = 3$ because the Max-$k$-Cut problem cannot be modeled as a QUBO problem. Furthermore, its heuristic rounding lacks theoretical guarantees, which results in sub-optimal performance regarding objective function values. Traditional methods such as BQP

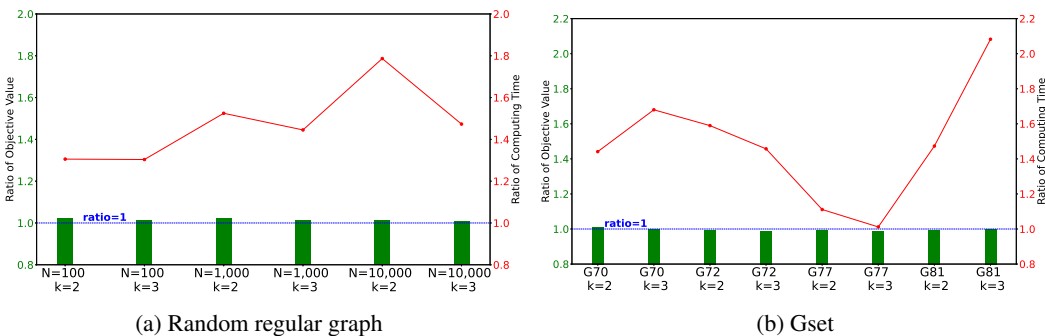

Figure 3: The ratio of computational time and objective value comparison of Max-$k$-Cut problems between `ROS-vanilla` and `ROS`.

and `Genetic` can accommodate $k = 3$, but they often become trapped in sub-optimal solutions. Among all the baselines, only `MD` can handle general $k$ while producing solutions of comparable quality to `ROS`. However, `MD` consistently exhibits inferior performance compared to `ROS` across all experiments. The results for the Gset benchmark, shown in Table 2, offer similar insights: `ROS` demonstrates better generalizability compared to the traditional `Rank2` method and the L2O method `PI-GNN`. Moreover, `ROS` yields higher-quality solutions than `MD` in terms of objective function values.

### 4.3 EFFECT OF THE "PRE-TRAIN" STAGE IN `ROS`

To evaluate the impact of the pre-training stage in `ROS`, we compared it with `ROS-vanilla`, a variant that omits the pre-training phase (see Section 3.3). We assessed both methods based on objective function values and computational time. Figure 3 illustrates the ratios of these metrics between `ROS-vanilla` and `ROS`. In this figure, the horizontal axis represents the problem instances, while the left vertical axis (green bars) displays the ratio of objective function values, and the right vertical axis (red curve) indicates the ratio of computational times.

As shown in Figure 3a, during experiments on regular graphs, `ROS-vanilla` achieves higher objective function values in most settings; however, its computational time is approximately 1.5 times greater than that of `ROS`. Thus, `ROS` demonstrates a faster solving speed compared to `ROS-vanilla`. Similarly, in experiments conducted on the Gset benchmark (Figure 3b), `ROS` reduces computational time by around 40% while maintaining performance comparable to that of `ROS-vanilla`. Notably, in the Max-3-Cut problem for the largest instance, G81, `ROS` effectively halves the solving time, showcasing the significant acceleration effect of pre-training. It is worth mentioning that the `ROS` model was pre-trained on random regular graphs with $N = 100$ and generalized well to regular graphs with $N = 1,000$ and $N = 10,000$, as well as to Gset problem instances of varying sizes and types. This illustrates `ROS`'s capability to generalize and accelerate the solving of large-scale problems across diverse graph types and sizes, emphasizing the strong out-of-distribution generalization afforded by pre-training.

In summary, while `ROS-vanilla` achieves slightly higher objective function values on individual instances, it requires longer solving times and struggles to generalize to other problem instances. This observation highlights the trade-off between a model's ability to generalize and its capacity to fit specific instances. Specifically, a model that fits individual instances exceptionally well may fail to generalize to new data, resulting in longer solving times. Conversely, a model that generalizes effectively may exhibit slightly weaker performance on specific instances, leading to a marginal decrease in objective function values. Therefore, the choice between these two training modes should be guided by the specific requirements of the application.

## 5 CONCLUSIONS

In this paper, we propose ROS, an efficient method for addressing the Max-$k$-Cut problem with any arbitrary edge weights. Our approach begins by relaxing the constraints of the original dis-

crete problem to probabilistic simplices. To effectively solve this relaxed problem, we propose an optimization algorithm based on GNN parametrization and incorporate transfer learning by leveraging pre-trained GNNs to warm-start the training process. After resolving the relaxed problem, we present a novel random sampling algorithm that maps the continuous solution back to a discrete form. By integrating geometric landscape analysis with statistical theory, we establish the consistency of function values between the continuous and discrete solutions. Experiments conducted on random regular graphs and the Gset benchmark demonstrate that our method is highly efficient for solving large-scale Max-$k$-Cut problems, requiring only a few seconds, even for instances with tens of thousands of variables. Furthermore, it exhibits robust generalization capabilities across both in-distribution and out-of-distribution instances, highlighting its effectiveness for large-scale optimization tasks. Exploring other sampling algorithms to further boost ROS performance is a future research direction. Moreover, the ROS framework with theoretical insights could be potentially extended to other graph-related combinatorial problems, and this direction is also worth investigating as future work.

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

## A  PROOF OF THEOREM 1

*Proof.* Before proceeding with the proof of Theorem 1, we first define the neighborhood of a vector $\bar{\boldsymbol{x}} \in \Delta_k$, and establish results of Lemma 1 and Lemma 2.

**Definition 2.** *Let $\bar{\boldsymbol{x}} = (\bar{\boldsymbol{x}}_1, \cdots, \bar{\boldsymbol{x}}_k)$ denote a point in $\Delta_k$. We define the neighborhood induced by $\bar{\boldsymbol{x}}$ as follows:*

$$
\widetilde{\mathcal{N}}(\bar{\boldsymbol{x}}) := \left\{ (\boldsymbol{x}_1, \cdots, \boldsymbol{x}_k) \in \Delta_k \ \middle| \ \sum_{j \in \mathcal{K}(\bar{\boldsymbol{x}})} \boldsymbol{x}_j = 1 \right\},
$$

*where $\mathcal{K}(\bar{\boldsymbol{x}}) = \{j \in \{1, \cdots, k\} \mid \bar{\boldsymbol{x}}_j > 0\}$.*

**Lemma 1.** *Given $\boldsymbol{X}_{\cdot i} \in \widetilde{\mathcal{N}}(\overline{\boldsymbol{X}}_{\cdot i})$, it follows that*

$$
\mathcal{K}(\boldsymbol{X}_{\cdot i}) \subseteq \mathcal{K}(\overline{\boldsymbol{X}}_{\cdot i}).
$$

*Proof.* Suppose there exists $j \in \mathcal{K}(\boldsymbol{X}_{\cdot i})$ such that $j \notin \mathcal{K}(\overline{\boldsymbol{X}}_{\cdot i})$, implying $\boldsymbol{X}_{ji} > 0$ and $\overline{\boldsymbol{X}}_{ji} = 0$. We then have

$$
\sum_{l \in \mathcal{K}(\overline{\boldsymbol{X}}_{\cdot i})} \boldsymbol{X}_{li} + \boldsymbol{X}_{ji} \leq \sum_{l=1}^{k} \boldsymbol{X}_{li} = 1,
$$

which leads to

$$
\sum_{l \in \mathcal{K}(\overline{\boldsymbol{X}}_{\cdot i})} \boldsymbol{X}_{li} \leq 1 - \boldsymbol{X}_{ji} < 1,
$$

contradicting with the fact that $\boldsymbol{X}_{\cdot i} \in \widetilde{\mathcal{N}}(\overline{\boldsymbol{X}}_{\cdot i})$. $\qquad\square$

**Lemma 2.** *Let $\overline{\boldsymbol{X}}$ be a globally optimal solution to $\overline{\boldsymbol{P}}$, then*

$$
f(\boldsymbol{X}; \boldsymbol{W}) = f(\overline{\boldsymbol{X}}; \boldsymbol{W}),
$$

*where $\boldsymbol{X}$ has only the $i^{th}$ column $\boldsymbol{X}_{\cdot i} \in \widetilde{\mathcal{N}}(\overline{\boldsymbol{X}}_{\cdot i})$, and other columns are identical to those of $\overline{\boldsymbol{X}}$. Moreover, $\boldsymbol{X}$ is also a globally optimal solution to $\overline{\boldsymbol{P}}$.*

*Proof.* The fact that $\boldsymbol{X}$ is a globally optimal solution to $\overline{\boldsymbol{P}}$ follows directly from the equality $f(\boldsymbol{X}; \boldsymbol{W}) = f(\overline{\boldsymbol{X}}; \boldsymbol{W})$. Thus, it suffices to prove this equality. Consider that $\overline{\boldsymbol{X}}$ and $\boldsymbol{X}$ differ only in the $i^{th}$ column, and $\boldsymbol{X}_{\cdot i} \in \widetilde{\mathcal{N}}(\overline{\boldsymbol{X}}_{\cdot i})$. We can rewrite the objective value function as

$$
f(\boldsymbol{X}; \boldsymbol{W}) = g(\boldsymbol{X}_{\cdot i}; \boldsymbol{X}_{\cdot -i}) + h(\boldsymbol{X}_{\cdot -i}),
$$

where $\boldsymbol{X}_{\cdot -i}$ represents all column vectors of $\boldsymbol{X}$ except the $i^{th}$ column. The functions $g$ and $h$ are defined as follows:

$$
g(\boldsymbol{X}_{\cdot i}; \boldsymbol{X}_{\cdot -i}) = \sum_{j=1}^{N} \boldsymbol{W}_{ij} \boldsymbol{X}_{\cdot i}^{\top} \boldsymbol{X}_{\cdot j} + \sum_{j=1}^{N} \boldsymbol{W}_{ji} \boldsymbol{X}_{\cdot j}^{\top} \boldsymbol{X}_{\cdot i} - \boldsymbol{W}_{ii} \boldsymbol{X}_{\cdot i}^{\top} \boldsymbol{X}_{\cdot i},
$$

$$
h(\boldsymbol{X}_{\cdot -i}) = \sum_{l=1, l \neq i}^{N} \sum_{j=1, j \neq i}^{N} \boldsymbol{W}_{lj} \boldsymbol{X}_{\cdot l}^{\top} \boldsymbol{X}_{\cdot j}
$$

To establish that $f(\boldsymbol{X}; \boldsymbol{W}) = f(\overline{\boldsymbol{X}}; \boldsymbol{W})$, it suffices to show that

$$
g(\boldsymbol{X}_{\cdot i}; \boldsymbol{X}_{\cdot -i}) = g(\overline{\boldsymbol{X}}_{\cdot i}; \boldsymbol{X}_{\cdot -i})
$$

as $\boldsymbol{X}_{\cdot -i} = \overline{\boldsymbol{X}}_{\cdot -i}$.

Rewriting $g(\boldsymbol{X}_{\cdot i}; \boldsymbol{X}_{\cdot -i})$, we obtain

$$g(\boldsymbol{X}_{\cdot i}; \boldsymbol{X}_{\cdot -i}) = \sum_{j=1}^{N} \boldsymbol{W}_{ij} \boldsymbol{X}_{\cdot i}^{\top} \boldsymbol{X}_{\cdot j} + \sum_{j=1}^{N} \boldsymbol{W}_{ji} \boldsymbol{X}_{\cdot j}^{\top} \boldsymbol{X}_{\cdot i}$$

$$= 2 \sum_{j=1}^{N} \boldsymbol{W}_{ij} \boldsymbol{X}_{\cdot i}^{\top} \boldsymbol{X}_{\cdot j}$$

$$= 2 \boldsymbol{X}_{\cdot i}^{\top} \sum_{j=1, j \neq i}^{N} \boldsymbol{W}_{ij} \boldsymbol{X}_{\cdot j}$$

$$= 2 \boldsymbol{X}_{\cdot i}^{\top} \boldsymbol{Y}_{\cdot i},$$

where $\boldsymbol{Y}_{\cdot i} := \sum_{j=1, j \neq i}^{N} \boldsymbol{W}_{ij} \boldsymbol{X}_{\cdot j}$.

If $|\mathcal{K}(\overline{\boldsymbol{X}}_{\cdot i})| = 1$, then there is only one non-zero element in $\overline{\boldsymbol{X}}_{\cdot i}$ equal to one. Therefore, $g(\overline{\boldsymbol{X}}_{\cdot i}; \boldsymbol{X}_{\cdot -i}) = g(\boldsymbol{X}_{\cdot i}; \boldsymbol{X}_{\cdot -i})$ since $\boldsymbol{X}_{\cdot i} = \overline{\boldsymbol{X}}_{\cdot i}$.

For the case where $|\mathcal{K}(\overline{\boldsymbol{X}}_{\cdot i})| > 1$, we consider any indices $j, l \in \mathcal{K}(\overline{\boldsymbol{X}}_{\cdot i})$ such that $\overline{\boldsymbol{X}}_{ji}, \overline{\boldsymbol{X}}_{li} > 0$. Then, there exists $\epsilon > 0$ such that we can construct a point $\widetilde{\boldsymbol{x}} \in \Delta_k$ where the $j^{th}$ element is set to $\overline{\boldsymbol{X}}_{ji} - \epsilon$, the $l^{th}$ element is set to $\overline{\boldsymbol{X}}_{li} + \epsilon$, and all other elements remain the same as in $\overline{\boldsymbol{X}}_{\cdot i}$. Since $\overline{\boldsymbol{X}}$ is a globally optimum of the function $f(\boldsymbol{X}; \boldsymbol{W})$, it follows that $\overline{\boldsymbol{X}}_{\cdot i}$ is also a global optimum for the function $g(\overline{\boldsymbol{X}}_{\cdot i}; \boldsymbol{X}_{\cdot -i})$. Thus, we have

$$g(\overline{\boldsymbol{X}}_{\cdot i}; \boldsymbol{X}_{\cdot -i}) \leq g(\widetilde{\boldsymbol{x}}; \boldsymbol{X}_{\cdot -i})$$

$$\overline{\boldsymbol{X}}_{\cdot i}^{\top} \boldsymbol{Y}_{\cdot i} \leq \widetilde{\boldsymbol{x}}^{\top} \boldsymbol{Y}_{\cdot i}$$

$$= \overline{\boldsymbol{X}}_{\cdot i}^{\top} \boldsymbol{Y}_{\cdot i} - \epsilon \boldsymbol{Y}_{ji} + \epsilon \boldsymbol{Y}_{li},$$

which leads to the inequality

$$\boldsymbol{Y}_{ji} \leq \boldsymbol{Y}_{li}. \tag{3}$$

Next, we can similarly construct another point $\hat{\boldsymbol{x}} \in \Delta_k$ with its $j^{th}$ element equal to $\overline{\boldsymbol{X}}_{ji} + \epsilon$, the $k^{th}$ element equal to $\overline{\boldsymbol{X}}_{ki} - \epsilon$, and all other elements remain the same as in $\overline{\boldsymbol{X}}_{\cdot i}$. Subsequently, we can also derive that

$$g(\overline{\boldsymbol{X}}_{\cdot i}; \boldsymbol{X}_{\cdot -i}) \leq g(\hat{\boldsymbol{x}}; \boldsymbol{X}_{\cdot -i})$$

$$= \overline{\boldsymbol{X}}_{\cdot i}^{\top} \boldsymbol{Y}_{\cdot i} + \epsilon \boldsymbol{Y}_{ji} - \epsilon \boldsymbol{Y}_{li},$$

which leads to another inequality

$$\boldsymbol{Y}_{li} \leq \boldsymbol{Y}_{ji}. \tag{4}$$

Consequently, combined inequalities (3) and (4), we have

$$\boldsymbol{Y}_{ji} = \boldsymbol{Y}_{li},$$

for $j, l \in \mathcal{K}(\overline{\boldsymbol{X}}_{\cdot i})$.

From this, we can deduce that

$$\boldsymbol{Y}_{j_1 i} = \boldsymbol{Y}_{j_2 i} = \cdots = \boldsymbol{Y}_{j_{|\mathcal{K}(\overline{\boldsymbol{X}}_{\cdot i})|} i} = t,$$

where $j_1, \cdots, j_{|\mathcal{K}(\overline{\boldsymbol{X}}_{\cdot i})|} \in \mathcal{K}(\overline{\boldsymbol{X}}_{\cdot i})$.

Next, we find that

$$
\begin{aligned}
g(\overline{\boldsymbol{X}}_{\cdot i}; \boldsymbol{X}_{\cdot -i}) &= 2\overline{\boldsymbol{X}}_{\cdot i}^\top \boldsymbol{Y}_{\cdot i} \\
&= 2\sum_{j=1}^{k} \overline{\boldsymbol{X}}_{ji}\boldsymbol{Y}_{ji} \\
&= 2\sum_{j=1, j\in\mathcal{K}(\overline{\boldsymbol{X}}_{\cdot i})}^{N} \overline{\boldsymbol{X}}_{ji}\boldsymbol{Y}_{ji} \\
&= 2t\sum_{j=1, j\in\mathcal{K}(\overline{\boldsymbol{X}}_{\cdot i})}^{N} \overline{\boldsymbol{X}}_{ji} \\
&= 2t.
\end{aligned}
$$

Similarly, we have

$$
\begin{aligned}
g(\boldsymbol{X}_{\cdot i}; \boldsymbol{X}_{\cdot -i}) &= 2\boldsymbol{X}_{\cdot i}^\top \boldsymbol{Y}_{\cdot i} \\
&= 2\sum_{j=1}^{k} \boldsymbol{X}_{ji}\boldsymbol{Y}_{ji} \\
&= 2\sum_{j=1, j\in\mathcal{K}(\boldsymbol{X}_{\cdot i})} \boldsymbol{X}_{ji}\boldsymbol{Y}_{ji} \\
&\overset{\text{Lemma 1}}{=} 2t\sum_{j=1, j\in\mathcal{K}(\boldsymbol{X}_{\cdot i})} \boldsymbol{X}_{ji} \\
&= 2t \\
&= g(\overline{\boldsymbol{X}}_{\cdot i})
\end{aligned}
$$

Accordingly, we conclude that

$$
g(\boldsymbol{X}_{\cdot i}; \boldsymbol{X}_{\cdot -i}) = g(\overline{\boldsymbol{X}}_{\cdot i}; \boldsymbol{X}_{\cdot -i}),
$$

which leads us to the result

$$
f(\boldsymbol{X}; \boldsymbol{W}) = f(\overline{\boldsymbol{X}}; \boldsymbol{W}),
$$

where $\boldsymbol{X}_{\cdot i} \in \widetilde{\mathcal{N}}(\overline{\boldsymbol{X}}_{\cdot i})$, $\boldsymbol{X}_{\cdot -i} = \overline{\boldsymbol{X}}_{\cdot -i}$. $\qquad\square$

Accordingly, for any $\boldsymbol{X} \in \mathcal{N}(\overline{\boldsymbol{X}})$, we iteratively apply Lemma 2 to each column of $\overline{\boldsymbol{X}}$ while holding the other columns fixed, thereby proving Theorem 1.

$\qquad\square$

## B  PROOF OF THEOREM 2

*Proof.* Based on $\overline{\boldsymbol{X}}$, we can construct the random variable $\widetilde{\boldsymbol{X}}$, where $\widetilde{\boldsymbol{X}}_{\cdot i} \sim \text{Cat}(\boldsymbol{x}; \boldsymbol{p} = \overline{\boldsymbol{X}}_{\cdot i})$. The probability mass function is given by

$$
\mathbf{P}(\widetilde{\boldsymbol{X}}_{\cdot i} = \boldsymbol{e}_\ell) = \overline{\boldsymbol{X}}_{\ell i}, \tag{5}
$$

where $\ell = 1, \cdots, k$.

Next, we have

$$\mathbb{E}_{\widetilde{\boldsymbol{X}}}[f(\widetilde{\boldsymbol{X}};\boldsymbol{W})] = \mathbb{E}_{\widetilde{\boldsymbol{X}}}[\widetilde{\boldsymbol{X}}\boldsymbol{W}\widetilde{\boldsymbol{X}}^\top] = \mathbb{E}_{\widetilde{\boldsymbol{X}}}[\sum_{i=1}^{N}\sum_{j=1}^{N}\boldsymbol{W}_{ij}\widetilde{\boldsymbol{X}}_{\cdot i}^\top\widetilde{\boldsymbol{X}}_{\cdot j}]$$

$$= \sum_{i=1}^{N}\sum_{j=1}^{N}\boldsymbol{W}_{ij}\mathbb{E}_{\widetilde{\boldsymbol{X}}_{\cdot i}\widetilde{\boldsymbol{X}}_{\cdot j}}[\widetilde{\boldsymbol{X}}_{\cdot i}^\top\widetilde{\boldsymbol{X}}_{\cdot j}]$$

$$= \sum_{i=1}^{N}\sum_{j=1}^{N}\boldsymbol{W}_{ij}\mathbb{E}_{\widetilde{\boldsymbol{X}}_{\cdot i}\widetilde{\boldsymbol{X}}_{\cdot j}}[\mathbb{1}(\widetilde{\boldsymbol{X}}_{\cdot i} = \widetilde{\boldsymbol{X}}_{\cdot j})]$$

$$= \sum_{i=1}^{N}\sum_{j=1}^{N}\boldsymbol{W}_{ij}\mathbb{P}(\widetilde{\boldsymbol{X}}_{\cdot i} = \widetilde{\boldsymbol{X}}_{\cdot j})$$

$$= \sum_{i=1}^{N}\sum_{j=1,j\neq i}^{N}\boldsymbol{W}_{ij}\mathbb{P}(\widetilde{\boldsymbol{X}}_{\cdot i} = \widetilde{\boldsymbol{X}}_{\cdot j}). \tag{6}$$

Since $\widetilde{\boldsymbol{X}}_{\cdot i}$ and $\widetilde{\boldsymbol{X}}_{\cdot j}$ are independent for $i \neq j$, we have

$$\mathbb{P}(\widetilde{\boldsymbol{X}}_{\cdot i} = \widetilde{\boldsymbol{X}}_{\cdot j}) = \sum_{\ell=1}^{k}\mathbb{P}(\widetilde{\boldsymbol{X}}_{\cdot i} = \widetilde{\boldsymbol{X}}_{\cdot j} = \boldsymbol{e}_\ell)$$

$$= \sum_{\ell=1}^{k}\mathbb{P}(\widetilde{\boldsymbol{X}}_{\cdot i} = \boldsymbol{e}_\ell, \widetilde{\boldsymbol{X}}_{\cdot j} = \boldsymbol{e}_\ell)$$

$$= \sum_{\ell=1}^{k}\mathbb{P}(\widetilde{\boldsymbol{X}}_{\cdot i} = \boldsymbol{e}_\ell)\mathbb{P}(\widetilde{\boldsymbol{X}}_{\cdot j} = \boldsymbol{e}_\ell)$$

$$= \sum_{\ell=1}^{k}\overline{\boldsymbol{X}}_{\ell i}\overline{\boldsymbol{X}}_{\ell j}$$

$$= \overline{\boldsymbol{X}}_{\cdot i}^\top\overline{\boldsymbol{X}}_{\cdot j}. \tag{7}$$

Substitute (7) into (6), we obtain

$$\mathbb{E}_{\widetilde{\boldsymbol{X}}}[f(\widetilde{\boldsymbol{X}};\boldsymbol{W})] = \sum_{i=1}^{N}\sum_{j=1}^{N}\boldsymbol{W}_{ij}\overline{\boldsymbol{X}}_{\cdot i}^\top\overline{\boldsymbol{X}}_{\cdot j} = f(\overline{\boldsymbol{X}};\boldsymbol{W}). \tag{8}$$

$\square$

## C  THE COMPLETE RESULTS ON GSET INSTANCES

Table 3: Complete results on Gset instances for Max-Cut. "★" indicates missing results from the literature.

| Instance | $|\mathcal{V}|$ | $|\mathcal{E}|$ | GW Obj. ↑ | GW Time (s) ↓ | MD Obj. ↑ | MD Time (s) ↓ | Rank2 Obj. ↑ | Rank2 Time (s) ↓ | PI-GNN Obj. ↑ | PI-GNN Time (s) ↓ | Genetic Obj. ↑ | Genetic Time (s) ↓ | BQP Obj. ↑ | BQP Time (s) ↓ | MOH Obj. ↑ | MOH Time (s) ↓ | LPI Obj. ↑ | LPI Time (s) ↓ | ROS-vanilla Obj. ↑ | ROS-vanilla Time (s) ↓ | ROS Obj. ↑ | ROS Time (s) ↓ |
|---|---|---|---|---|---|---|---|---|---|---|---|---|---|---|---|---|---|---|---|---|---|---|
| G1 | 800 | 19176 | 11299 | 1228.0 | 11320 | 5.1 | ★ | ★ | 11258 | 44.7 | 10929 | 587.4 | 11406 | 11.3 | 11624 | 1.5 | 11624 | 7 | 11423 | 2.6 | 11395 | 1.7 |
| G2 | 800 | 19176 | 11299 | 1225.4 | 11255 | 5.3 | ★ | ★ | 11258 | 45.6 | 10926 | 588.3 | 11426 | 11.7 | 11620 | 4.6 | 11620 | 8 | 11462 | 2.6 | 11467 | 1.8 |
| G3 | 800 | 19176 | 11289 | 1243.2 | 11222 | 5.3 | ★ | ★ | 11262 | 45.3 | 10933 | 596.8 | 11397 | 11.0 | 11622 | 1.3 | 11622 | 10 | 11510 | 2.7 | 11370 | 1.9 |
| G4 | 800 | 19176 | 11207 | 1217.8 | 11280 | 4.8 | ★ | ★ | 11216 | 44.9 | 10945 | 580.5 | 11430 | 11.2 | 11646 | 5.2 | 11646 | 7 | 11416 | 2.6 | 11459 | 2.1 |
| G5 | 800 | 19176 | 11256 | 1261.8 | 11156 | 3.7 | ★ | ★ | 11185 | 46.2 | 10869 | 598.2 | 11406 | 1.0 | 11631 | 1.0 | 11631 | 7 | 11505 | 2.6 | 11408 | 1.7 |
| G6 | 800 | 19176 | 1776 | 1261.6 | 1755 | 6.9 | ★ | ★ | 1418 | 201.4 | 1435 | 581.2 | 1991 | 11.4 | 2178 | 3.0 | 2178 | 14 | 1994 | 2.5 | 1907 | 1.7 |
| G7 | 800 | 19176 | 1694 | 1336.4 | 1635 | 5.9 | ★ | ★ | 1280 | 191.7 | 1273 | 587.5 | 1780 | 11.1 | 2006 | 3.0 | 2006 | 7 | 1802 | 2.6 | 1804 | 1.8 |
| G8 | 800 | 19176 | 1693 | 1235.2 | 1651 | 6.1 | ★ | ★ | 1285 | 201.0 | 1241 | 591.8 | 1758 | 11.1 | 2005 | 5.7 | 2005 | 10 | 1876 | 2.8 | 1775 | 1.8 |
| G9 | 800 | 19176 | 1676 | 1215.0 | 1720 | 8.0 | ★ | ★ | 1332 | 201.5 | 1345 | 582.3 | 1845 | 14.6 | 2054 | 3.2 | 2054 | 13 | 1839 | 2.6 | 1876 | 1.9 |
| G10 | 800 | 19176 | 1675 | 1227.3 | 1700 | 7.3 | ★ | ★ | 1299 | 201.4 | 1313 | 589.5 | 1816 | 10.9 | 2000 | 68.1 | 2000 | 10 | 1811 | 2.6 | 1755 | 1.8 |
| G11 | 800 | 1600 | N/A | N/A | 466 | 3.0 | 554 | 3.9 | 368 | 22.4 | 406 | 509.4 | 540 | 11.0 | 564 | 0.2 | 564 | 11 | 496 | 1.8 | 494 | 1.5 |
| G12 | 800 | 1600 | N/A | N/A | 466 | 2.4 | 552 | 3.8 | 386 | 21.8 | 388 | 514.8 | 534 | 11.0 | 556 | 3.5 | 556 | 16 | 498 | 1.9 | 494 | 1.4 |
| G13 | 800 | 1600 | N/A | N/A | 486 | 3.0 | 572 | 3.5 | 362 | 20.6 | 426 | 520.0 | 560 | 10.8 | 582 | 0.9 | 582 | 23 | 518 | 1.9 | 524 | 1.5 |
| G14 | 800 | 4694 | 2942 | 1716.6 | 2930 | 3.1 | 3053 | 5.5 | 2248 | 41.9 | 2855 | 564.2 | 2985 | 11.1 | 3064 | 251.3 | 3064 | 119 | 2932 | 1.5 | 2953 | 1.8 |
| G15 | 800 | 4661 | N/A | N/A | 2932 | 3.1 | 3039 | 5.9 | 2199 | 40.8 | 2836 | 547.7 | 2966 | 11.1 | 3050 | 52.2 | 3050 | 80 | 2920 | 1.8 | 2871 | 1.4 |
| G16 | 800 | 4672 | N/A | N/A | 2937 | 3.8 | ★ | ★ | 2359 | 50.8 | 2848 | 541.3 | 2987 | 14.3 | 3052 | 93.7 | 3052 | 69 | 2917 | 1.7 | 2916 | 1.3 |
| G17 | 800 | 4667 | 2916 | 1738.2 | 2922 | 3.3 | ★ | ★ | 2061 | 41.3 | 2829 | 558.9 | 2967 | 12.1 | 3047 | 129.5 | 3047 | 104 | 2932 | 1.9 | 2914 | 1.5 |
| G18 | 800 | 4694 | 838 | 871.7 | 825 | 3.7 | ★ | ★ | 596 | 34.9 | 643 | 567.0 | 922 | 12.1 | 992 | 112.7 | 992 | 40 | 903 | 2.1 | 905 | 1.7 |
| G19 | 800 | 4661 | 763 | 1245.4 | 740 | 3.6 | ★ | ★ | 528 | 31.1 | 571 | 571.2 | 816 | 11.4 | 906 | 266.9 | 906 | 49 | 808 | 2 | 772 | 1.5 |
| G20 | 800 | 4672 | 781 | 1015.6 | 767 | 3.5 | 939 | 5.6 | 592 | 33.8 | 633 | 565.8 | 860 | 11.9 | 941 | 43.7 | 941 | 31 | 843 | 2.1 | 788 | 1.8 |
| G21 | 800 | 4667 | 821 | 1350.3 | 784 | 3.0 | 921 | 5.6 | 617 | 32.4 | 620 | 572.2 | 837 | 14.1 | 931 | 155.3 | 931 | 32 | 858 | 2.1 | 848 | 1.6 |
| G22 | 2000 | 19990 | N/A | N/A | 12777 | 12.2 | 13331 | 22.3 | 12757 | 37.5 | N/A | N/A | 13004 | 95.6 | 13359 | 352.4 | 13359 | 413 | 13028 | 2.6 | 13007 | 2.7 |
| G23 | 2000 | 19990 | N/A | N/A | 12688 | 10.2 | 13269 | 18.9 | 12718 | 38.0 | N/A | N/A | 12958 | 95.6 | 13344 | 433.8 | 13342 | 150 | 13048 | 2.9 | 12936 | 1.9 |
| G24 | 2000 | 19990 | N/A | N/A | 12721 | 10.0 | 13287 | 27.3 | 12565 | 37.5 | N/A | N/A | 13002 | 95.0 | 13337 | 777.9 | 13337 | 234 | 13035 | 2.4 | 12933 | 2.4 |
| G25 | 2000 | 19990 | N/A | N/A | 12725 | 11.7 | ★ | ★ | 12617 | 37.9 | N/A | N/A | 12968 | 102.6 | 13340 | 442.5 | 13340 | 258 | 13040 | 2 | 12947 | 1.9 |
| G26 | 2000 | 19990 | N/A | N/A | 12725 | 10.8 | ★ | ★ | 12725 | 37.2 | N/A | N/A | 12966 | 96.9 | 13328 | 535.1 | 13328 | 291 | 13054 | 2.5 | 12954 | 3.5 |
| G27 | 2000 | 19990 | N/A | N/A | 2632 | 11.2 | ★ | ★ | 2234 | 56.8 | N/A | N/A | 3062 | 98.9 | 3341 | 42.3 | 3341 | 152 | 2993 | 2.8 | 2971 | 2.1 |
| G28 | 2000 | 19990 | N/A | N/A | 2762 | 11.2 | ★ | ★ | 2069 | 58.1 | N/A | N/A | 2963 | 96.8 | 3298 | 707.2 | 3298 | 197 | 2985 | 2.6 | 2923 | 1.9 |
| G29 | 2000 | 19990 | N/A | N/A | 2736 | 12.3 | ★ | ★ | 2158 | 71.2 | N/A | N/A | 3044 | 96.4 | 3405 | 555.2 | 3405 | 293 | 3056 | 2.9 | 3089 | 2.9 |
| G30 | 2000 | 19990 | N/A | N/A | 2774 | 11.7 | 3377 | 23.8 | 2234 | 52.6 | N/A | N/A | 3074 | 99.3 | 3413 | 330.5 | 3413 | 410 | 3004 | 2.8 | 3025 | 2.9 |
| G31 | 2000 | 19990 | N/A | N/A | 2736 | 11.5 | 3255 | 19.6 | 2208 | 81.4 | N/A | N/A | 2998 | 96.3 | 3310 | 592.6 | 3310 | 412 | 3015 | 2.1 | 2943 | 1.9 |
| G32 | 2000 | 4000 | N/A | N/A | 1136 | 6.8 | 1380 | 13.1 | 956 | 30.7 | N/A | N/A | 1338 | 92.7 | 1410 | 65.8 | 1410 | 330 | 1240 | 2.2 | 1226 | 1.7 |
| G33 | 2000 | 4000 | N/A | N/A | 1106 | 6.6 | 1352 | 12.6 | 880 | 33.7 | N/A | N/A | 1302 | 89.3 | 1382 | 504.1 | 1382 | 349 | 1224 | 2.3 | 1208 | 1.7 |
| G34 | 2000 | 4000 | N/A | N/A | 1118 | 5.8 | 1358 | 9.8 | 912 | 32.6 | N/A | N/A | 1314 | 95.6 | 1384 | 84.2 | 1384 | 302 | 1238 | 2.3 | 1220 | 1.6 |
| G35 | 2000 | 11778 | N/A | N/A | 7358 | 9.4 | ★ | ★ | 5574 | 39.5 | N/A | N/A | 7495 | 95.2 | 7686 | 796.7 | 7686 | 1070 | 7245 | 1.9 | 7260 | 1.9 |

Table 3: Continued.

| Instance | \|V\| | \|E\| | GW Obj.↑ | GW Time(s)↓ | MD Obj.↑ | MD Time(s)↓ | Rank2 Obj.↑ | Rank2 Time(s)↓ | PI-GNN Obj.↑ | PI-GNN Time(s)↓ | Genetic Obj.↑ | Genetic Time(s)↓ | BQP Obj.↑ | BQP Time(s)↓ | MOH Obj.↑ | MOH Time(s)↓ | LPI Obj.↑ | LPI Time(s)↓ | ROS-vanilla Obj.↑ | ROS-vanilla Time(s)↓ | ROS Obj.↑ | ROS Time(s)↓ |
|---|---|---|---|---|---|---|---|---|---|---|---|---|---|---|---|---|---|---|---|---|---|---|
| G36 | 2000 | 11766 | N/A | N/A | 7336 | 10.1 | * | * | 5596 | 36.5 | N/A | N/A | 7490 | 95.3 | 7680 | 664.5 | 7680 | 5790 | 7235 | 2.4 | 7107 | 1.5 |
| G37 | 2000 | 11785 | N/A | N/A | 7400 | 9.3 | * | * | 6092 | 37.1 | N/A | N/A | 7498 | 95.4 | 7691 | 652.8 | 7691 | 4082 | 7164 | 1.7 | 7141 | 1.5 |
| G38 | 2000 | 11779 | N/A | N/A | 7343 | 8.6 | * | * | 5982 | 38.1 | N/A | N/A | 7507 | 100.6 | 7688 | 779.7 | 7688 | 614 | 7114 | 1.6 | 7173 | 1.8 |
| G39 | 2000 | 11778 | N/A | N/A | 1998 | 9.2 | * | * | 1461 | 201.5 | N/A | N/A | 2196 | 94.4 | 2408 | 787.7 | 2408 | 347 | 2107 | 2.5 | 2165 | 1.7 |
| G40 | 2000 | 11766 | N/A | N/A | 1971 | 9.0 | * | * | 1435 | 201.0 | N/A | N/A | 2169 | 97.3 | 2400 | 472.5 | 2400 | 286 | 2120 | 2.7 | 2128 | 2.5 |
| G41 | 2000 | 11785 | N/A | N/A | 1969 | 9.1 | * | * | 1478 | 105.5 | N/A | N/A | 2183 | 105.8 | 2405 | 377.4 | 2405 | 328 | 2120 | 1.6 | 2139 | 2.2 |
| G42 | 2000 | 11779 | N/A | N/A | 2075 | 9.5 | * | * | 1508 | 201.6 | N/A | N/A | 2255 | 95.5 | 2481 | 777.4 | 2481 | 20 | 2200 | 2.2 | 2235 | 2.4 |
| G43 | 1000 | 9990 | 6340 | 1784.5 | 6380 | 5.0 | * | * | 6434 | 40.9 | 5976 | 914.4 | 6509 | 18.0 | 6650 | 1.2 | 6650 | 19 | 6539 | 2.7 | 6471 | 1.7 |
| G44 | 1000 | 9990 | 6351 | 1486.7 | 6327 | 5.0 | * | * | 6367 | 40.8 | 6009 | 914.3 | 6463 | 18.5 | 6654 | 5.3 | 6654 | 20 | 6498 | 2.5 | 6472 | 1.7 |
| G45 | 1000 | 9990 | 6355 | 1582.0 | 6329 | 4.9 | * | * | 6341 | 41.6 | 6006 | 921.5 | 6489 | 22.4 | 6649 | 6.9 | 6649 | 19 | 6528 | 2.4 | 6489 | 2.5 |
| G46 | 1000 | 9990 | 6357 | 1612.8 | 6300 | 4.8 | * | * | 6312 | 41.1 | 5978 | 916.2 | 6485 | 18.4 | 6657 | 67.3 | 6657 | 21 | 6498 | 2.5 | 6499 | 1.8 |
| G47 | 1000 | 9990 | N/A | N/A | 6369 | 4.7 | * | * | 6391 | 40.4 | 5948 | 912.4 | 6491 | 18.4 | 6657 | 43.3 | 6657 | 25 | 6497 | 2.5 | 6489 | 2.1 |
| G48 | 3000 | 6000 | N/A | N/A | 5006 | 10.6 | 6000 | 13.1 | 5402 | 30.7 | N/A | N/A | 6000 | 300.4 | 6000 | 0.0 | 6000 | 94 | 5640 | 3.2 | 5498 | 2.2 |
| G49 | 3000 | 6000 | N/A | N/A | 5086 | 10.1 | 6000 | 11.4 | 5434 | 30.5 | N/A | N/A | 6000 | 303.0 | 6000 | 0.0 | 6000 | 93 | 5580 | 3.1 | 5452 | 1.9 |
| G50 | 3000 | 6000 | N/A | N/A | 5156 | 11.3 | 5856 | 15.7 | 5458 | 30.0 | N/A | N/A | 5880 | 299.8 | 5880 | 532.1 | 5880 | 90 | 5656 | 3.2 | 5582 | 1.7 |
| G51 | 1000 | 5909 | N/A | N/A | 3693 | 4.1 | * | * | 2841 | 40.6 | 3568 | 887.9 | 3759 | 17.7 | 3848 | 189.2 | 3848 | 145 | 3629 | 1.5 | 3677 | 1.6 |
| G52 | 1000 | 5916 | N/A | N/A | 3695 | 4.7 | * | * | 2615 | 41.2 | 3575 | 897.7 | 3771 | 18.5 | 3851 | 209.7 | 3851 | 119 | 3526 | 1.3 | 3641 | 1.6 |
| G53 | 1000 | 5914 | N/A | N/A | 3670 | 4.5 | * | * | 2813 | 41.1 | 3545 | 872.8 | 3752 | 18.0 | 3850 | 299.3 | 3850 | 182 | 3633 | 1.5 | 3658 | 1.3 |
| G54 | 1000 | 5916 | N/A | N/A | 3682 | 4.4 | * | * | 2790 | 41.3 | 3548 | 880.1 | 3753 | 18.0 | 3852 | 190.4 | 3852 | 140 | 3653 | 1.6 | 3642 | 2.9 |
| G55 | 5000 | 12498 | N/A | N/A | 9462 | 24.4 | 10240 | 39.7 | 9678 | 31.9 | N/A | N/A | 9862 | 1142.1 | 10299 | 1230.4 | 10299 | 6594 | 9819 | 2.1 | 9779 | 2.5 |
| G56 | 5000 | 12498 | N/A | N/A | 3203 | 23.8 | 3943 | 33.5 | 2754 | 217.2 | N/A | N/A | 3710 | 1147.6 | 4016 | 990.4 | 4017 | 49445 | 3444 | 2 | 3475 | 2.5 |
| G57 | 5000 | 10000 | N/A | N/A | 2770 | 17.3 | 3412 | 32.2 | 2266 | 218.4 | N/A | N/A | 3310 | 1120.8 | 3494 | 1528.3 | 3494 | 3494 | 3040 | 1.7 | 3078 | 1.8 |
| G58 | 5000 | 29570 | N/A | N/A | 18452 | 29.2 | * | * | 14607 | 39.7 | N/A | N/A | 18813 | 1176.6 | 19288 | 1522.3 | 19294 | 65737 | 17632 | 2.3 | 17574 | 4.7 |
| G59 | 5000 | 29570 | N/A | N/A | 5099 | 31.6 | * | * | 3753 | 216.8 | N/A | N/A | 5490 | 1183.4 | 6087 | 2498.8 | 6088 | 65112 | 5343 | 1.9 | 5407 | 2 |
| G60 | 7000 | 17148 | N/A | N/A | 13004 | 34.8 | 14081 | 57 | 13257 | 34.0 | N/A | N/A | N/A | N/A | 14190 | 2945.4 | 14190 | 44802 | 13433 | 2 | 13402 | 2 |
| G61 | 7000 | 17148 | N/A | N/A | 4592 | 36.0 | 5690 | 64 | 3963 | 233.0 | N/A | N/A | N/A | N/A | 5798 | 6603.3 | 5798 | 74373 | 5037 | 3.8 | 5011 | 2.8 |
| G62 | 7000 | 14000 | N/A | N/A | 3922 | 26.1 | 4740 | 47 | 3150 | 229.4 | N/A | N/A | N/A | N/A | 4868 | 5568.6 | 4872 | 26537 | 4252 | 1.7 | 4294 | 1.5 |
| G63 | 7000 | 41459 | N/A | N/A | 25938 | 45.1 | * | * | 19616 | 38.0 | N/A | N/A | N/A | N/A | 27033 | 6492.1 | 27033 | 52726 | 24185 | 2.3 | 24270 | 3 |
| G64 | 7000 | 41459 | N/A | N/A | 7283 | 43.7 | 8575 | 67.6 | 5491 | 205.8 | N/A | N/A | N/A | N/A | 8747 | 4011.1 | 8752 | 49158 | 7508 | 4.4 | 7657 | 2.5 |
| G65 | 8000 | 16000 | N/A | N/A | 4520 | 32.5 | * | * | 3680 | 232.8 | N/A | N/A | N/A | N/A | 5560 | 4709.5 | 5562 | 21737 | 4878 | 5.5 | 4826 | 3.3 |
| G66 | 9000 | 18000 | N/A | N/A | 5100 | 37.3 | * | * | 4112 | 241.3 | N/A | N/A | N/A | N/A | 6360 | 6061.9 | 6364 | 34062 | 5570 | 6.2 | 5580 | 1.9 |
| G67 | 10000 | 20000 | N/A | N/A | 5592 | 43.4 | 9529 | 94.4 | 4494 | 252.3 | N/A | N/A | N/A | N/A | 6942 | 4214.3 | 6948 | 61556 | 6090 | 4.9 | 6010 | 3.4 |
| G70 | 10000 | 9999 | N/A | N/A | 8551 | 54.3 | 6820 | 86.6 | 8956 | 34.5 | N/A | N/A | N/A | N/A | 9544 | 8732.4 | 9594 | 28820 | 9004 | 6.2 | 8916 | 3.9 |
| G72 | 10000 | 20000 | N/A | N/A | 5638 | 44.2 | 9670 | 109.4 | 4544 | 253.0 | N/A | N/A | N/A | N/A | 6998 | 6586.6 | 7004 | 42542 | 6066 | 9 | 6102 | 8.1 |
| G77 | 14000 | 28000 | N/A | N/A | 7934 | 66.0 | * | * | 6406 | 349.4 | N/A | N/A | N/A | N/A | 9928 | 9863.6 | 9926 | 66662 | 8678 | 13.7 | 8740 | 9.3 |
| G81 | 20000 | 40000 | N/A | N/A | 11226 | 130.8 | 13662 | 140.5 | 8970 | 557.7 | N/A | N/A | N/A | N/A | 14036 | 20422.0 | 14030 | 66691 | 12260 | — | 12332 | — |

Table 4: Complete results on Gset instances for Max-3-Cut.

| Instance | $|\mathcal{V}|$ | $|\mathcal{E}|$ | MD Obj. ↑ | MD Time (s) ↓ | Genetic Obj. ↑ | Genetic Time (s) ↓ | BQP Obj. ↑ | BQP Time (s) ↓ | MOH Obj. ↑ | MOH Time (s) ↓ | ROS-vanilla Obj. ↑ | ROS-vanilla Time (s) ↓ | ROS Obj. ↑ | ROS Time (s) ↓ |
|---|---|---|---|---|---|---|---|---|---|---|---|---|---|---|
| G1 | 800 | 19176 | 14735 | 9.6 | 14075 | 595.3 | 14880 | 16.5 | 15165 | 557.3 | 14949 | 2.8 | 14961 | 1.9 |
| G2 | 800 | 19176 | 14787 | 8.4 | 14035 | 595.3 | 14845 | 17.0 | 15172 | 333.3 | 15033 | 2.8 | 14932 | 2.3 |
| G3 | 800 | 19176 | 14663 | 6.5 | 14105 | 588.6 | 14872 | 17.0 | 15173 | 269.6 | 15016 | 2.9 | 14914 | 1.9 |
| G4 | 800 | 19176 | 14716 | 6.9 | 14055 | 588.7 | 14886 | 17.1 | 15184 | 300.6 | 14984 | 3.3 | 14961 | 1.9 |
| G5 | 800 | 19176 | 14681 | 8.1 | 14104 | 591.9 | 14847 | 17.3 | 15193 | 98.2 | 15006 | 3.2 | 14962 | 2.9 |
| G6 | 800 | 19176 | 2161 | 7.8 | 1504 | 604.4 | 2302 | 25.0 | 2632 | 307.3 | 2436 | 2.8 | 2361 | 1.8 |
| G7 | 800 | 19176 | 2017 | 8.9 | 1260 | 589.9 | 2081 | 16.6 | 2409 | 381.0 | 2188 | 2.1 | 2188 | 2.4 |
| G8 | 800 | 19176 | 1938 | 7.7 | 1252 | 589.7 | 2096 | 19.3 | 2428 | 456.5 | 2237 | 2.8 | 2171 | 2.1 |
| G9 | 800 | 19176 | 2031 | 8.2 | 1326 | 604.4 | 2099 | 16.5 | 2478 | 282.0 | 2246 | 2.8 | 2185 | 2.2 |
| G10 | 800 | 19176 | 1961 | 7.5 | 1266 | 593.3 | 2055 | 18.2 | 2407 | 569.3 | 2201 | 2.9 | 2181 | 2.3 |
| G11 | 800 | 1600 | 553 | 4.0 | 414 | 554.5 | 624 | 16.4 | 669 | 143.8 | 616 | 2 | 591 | 1.4 |
| G12 | 800 | 1600 | 530 | 4.4 | 388 | 543.6 | 608 | 17.4 | 660 | 100.7 | 604 | 2 | 582 | 1.5 |
| G13 | 800 | 1600 | 558 | 4.0 | 425 | 550.8 | 638 | 18.9 | 686 | 459.4 | 617 | 2 | 629 | 1.4 |
| G14 | 800 | 4694 | 3844 | 5.0 | 3679 | 571.1 | 3900 | 16.9 | 4012 | 88.2 | 3914 | 2.8 | 3892 | 2.1 |
| G15 | 800 | 4661 | 3815 | 4.8 | 3625 | 567.6 | 3885 | 17.3 | 3984 | 80.3 | 3817 | 1.9 | 3838 | 2 |
| G16 | 800 | 4672 | 3825 | 5.3 | 3642 | 561.5 | 3896 | 18.2 | 3991 | 1.3 | 3843 | 2.3 | 3845 | 1.6 |
| G17 | 800 | 4667 | 3815 | 5.3 | 3640 | 558.7 | 3886 | 20.2 | 3983 | 7.8 | 3841 | 2.4 | 3852 | 1.6 |
| G18 | 800 | 4694 | 992 | 4.5 | 704 | 584.0 | 1083 | 18.7 | 1207 | 0.3 | 1094 | 2.2 | 1067 | 1.7 |
| G19 | 800 | 4661 | 869 | 4.4 | 595 | 584.2 | 962 | 17.0 | 1081 | 0.2 | 972 | 2.1 | 967 | 1.7 |
| G20 | 800 | 4672 | 928 | 4.5 | 589 | 576.8 | 977 | 17.0 | 1122 | 13.3 | 1006 | 2.2 | 993 | 1.8 |
| G21 | 800 | 4667 | 936 | 4.9 | 612 | 576.3 | 984 | 17.5 | 1109 | 55.8 | 1011 | 2.2 | 975 | 1.5 |
| G22 | 2000 | 19990 | 16402 | 15.2 | N/A | N/A | 16599 | 135.5 | 17167 | 28.5 | 16790 | 3.3 | 16601 | 2.2 |
| G23 | 2000 | 19990 | 16422 | 15.0 | N/A | N/A | 16626 | 135.6 | 17168 | 45.1 | 16819 | 3.9 | 16702 | 2.1 |
| G24 | 2000 | 19990 | 16452 | 16.1 | N/A | N/A | 16591 | 137.7 | 17162 | 16.3 | 16801 | 3.6 | 16754 | 3 |
| G25 | 2000 | 19990 | 16407 | 16.2 | N/A | N/A | 16661 | 141.8 | 17163 | 64.8 | 16795 | 2.1 | 16673 | 1.8 |
| G26 | 2000 | 19990 | 16422 | 15.3 | N/A | N/A | 16608 | 136.3 | 17154 | 44.8 | 16758 | 3.1 | 16665 | 2 |
| G27 | 2000 | 19990 | 3250 | 16.4 | N/A | N/A | 3475 | 134.3 | 4020 | 53.2 | 3517 | 1.7 | 3532 | 2 |
| G28 | 2000 | 19990 | 3198 | 16.1 | N/A | N/A | 3433 | 136.4 | 3973 | 38.9 | 3507 | 3 | 3414 | 2.1 |
| G29 | 2000 | 19990 | 3324 | 16.0 | N/A | N/A | 3582 | 136.2 | 4106 | 68.2 | 3634 | 3.4 | 3596 | 2 |
| G30 | 2000 | 19990 | 3320 | 16.2 | N/A | N/A | 3578 | 133.6 | 4119 | 150.4 | 3656 | 3.1 | 3654 | 3.4 |
| G31 | 2000 | 19990 | 3243 | 17.0 | N/A | N/A | 3439 | 131.0 | 4003 | 124.7 | 3596 | 3 | 3525 | 2.5 |
| G32 | 2000 | 4000 | 1342 | 11.1 | N/A | N/A | 1545 | 129.3 | 1653 | 160.1 | 1488 | 2.5 | 1482 | 1.7 |
| G33 | 2000 | 4000 | 1284 | 10.7 | N/A | N/A | 1517 | 126.2 | 1625 | 62.6 | 1449 | 2.5 | 1454 | 2 |
| G34 | 2000 | 4000 | 1292 | 10.9 | N/A | N/A | 1499 | 126.0 | 1607 | 88.9 | 1418 | 2.4 | 1435 | 1.7 |
| G35 | 2000 | 11778 | 9644 | 14.2 | N/A | N/A | 9816 | 138.1 | 10046 | 66.2 | 9225 | 2 | 9536 | 1.7 |

Table 4: Continued.

| Instance | $|\mathcal{V}|$ | $|\mathcal{E}|$ | MD Obj. ↑ | MD Time (s) ↓ | Genetic Obj. ↑ | Genetic Time (s) ↓ | BQP Obj. ↑ | BQP Time (s) ↓ | MOH Obj. ↑ | MOH Time (s) ↓ | ROS-vanilla Obj. ↑ | ROS-vanilla Time (s) ↓ | ROS Obj. ↑ | ROS Time (s) ↓ |
|---|---|---|---|---|---|---|---|---|---|---|---|---|---|---|
| G36 | 2000 | 11766 | 9600 | 13.6 | N/A | N/A | 9786 | 138.6 | 10039 | 74.3 | 9372 | 2.1 | 9581 | 2.3 |
| G37 | 2000 | 11785 | 9632 | 14.9 | N/A | N/A | 9821 | 139.2 | 10052 | 3.4 | 8893 | 1.4 | 9422 | 1.5 |
| G38 | 2000 | 11779 | 9629 | 14.0 | N/A | N/A | 9775 | 142.3 | 10040 | 116.6 | 9489 | 2.5 | 9370 | 1.5 |
| G39 | 2000 | 11778 | 2368 | 13.4 | N/A | N/A | 2600 | 132.8 | 2903 | 9.0 | 2621 | 2.5 | 2557 | 2.2 |
| G40 | 2000 | 11766 | 2315 | 13.3 | N/A | N/A | 2568 | 131.2 | 2870 | 82.8 | 2474 | 2 | 2524 | 2.4 |
| G41 | 2000 | 11785 | 2386 | 12.7 | N/A | N/A | 2606 | 129.9 | 2887 | 87.7 | 2521 | 3.2 | 2584 | 2.5 |
| G42 | 2000 | 11779 | 2490 | 13.1 | N/A | N/A | 2682 | 129.2 | 2980 | 2.5 | 2638 | 2.7 | 2613 | 2.2 |
| G43 | 1000 | 9990 | 8214 | 8.1 | 7624 | 926.7 | 8329 | 29.9 | 8573 | 380.3 | 8414 | 2.6 | 8349 | 2.3 |
| G44 | 1000 | 9990 | 8187 | 7.0 | 7617 | 919.0 | 8326 | 27.7 | 8571 | 616.8 | 8369 | 2.6 | 8311 | 1.7 |
| G45 | 1000 | 9990 | 8226 | 7.7 | 7602 | 926.7 | 8296 | 34.2 | 8566 | 186.2 | 8397 | 2.9 | 8342 | 1.8 |
| G46 | 1000 | 9990 | 8229 | 7.5 | 7635 | 918.7 | 8312 | 27.8 | 8568 | 215.3 | 8409 | 2.6 | 8339 | 1.7 |
| G47 | 1000 | 9990 | 8211 | 7.2 | 7619 | 928.0 | 8322 | 27.3 | 8572 | 239.4 | 8386 | 2.6 | 8357 | 2.2 |
| G48 | 3000 | 6000 | 5806 | 14.7 | N/A | N/A | 5998 | 394.8 | 6000 | 0.4 | 5954 | 2.8 | 5912 | 2 |
| G49 | 3000 | 6000 | 5794 | 14.4 | N/A | N/A | 5998 | 404.0 | 6000 | 0.9 | 5938 | 2.8 | 5914 | 1.8 |
| G50 | 3000 | 6000 | 5823 | 14.5 | N/A | N/A | 6000 | 427.1 | 6000 | 119.2 | 5938 | 2.9 | 5918 | 1.8 |
| G51 | 1000 | 5909 | 4805 | 6.6 | 4582 | 889.5 | 4922 | 28.6 | 5037 | 47.9 | 4814 | 2.4 | 4820 | 1.7 |
| G52 | 1000 | 5916 | 4849 | 6.4 | 4571 | 908.1 | 4910 | 27.8 | 5040 | 0.7 | 4796 | 1.9 | 4866 | 1.9 |
| G53 | 1000 | 5914 | 4845 | 6.8 | 4568 | 898.6 | 4920 | 27.6 | 5039 | 223.9 | 4846 | 2.6 | 4808 | 1.6 |
| G54 | 1000 | 5916 | 4836 | 6.4 | 4562 | 911.7 | 4921 | 30.1 | 5036 | 134.0 | 4833 | 2.2 | 4785 | 1.4 |
| G55 | 5000 | 12498 | 11612 | 37.9 | N/A | N/A | 12042 | 1506.0 | 12429 | 383.1 | 12010 | 2.1 | 11965 | 2.6 |
| G56 | 5000 | 12498 | 3716 | 38.5 | N/A | N/A | 4205 | 1341.5 | 4752 | 569.2 | 4085 | 3.3 | 4037 | 2.1 |
| G57 | 5000 | 10000 | 3246 | 33.0 | N/A | N/A | 3817 | 1317.2 | 4083 | 535.6 | 3597 | 3.3 | 3595 | 2.8 |
| G58 | 5000 | 29570 | 24099 | 47.1 | N/A | N/A | 24603 | 1468.3 | 25195 | 576.0 | 22748 | 2.1 | 23274 | 1.9 |
| G59 | 5000 | 29570 | 6057 | 46.3 | N/A | N/A | 6631 | 1377.1 | 7262 | 27.5 | 6133 | 1.7 | 6448 | 3.5 |
| G60 | 7000 | 17148 | 15993 | 58.5 | N/A | N/A | N/A | N/A | 17076 | 683.0 | 16467 | 2.6 | 16398 | 2.3 |
| G61 | 7000 | 17148 | 5374 | 57.7 | N/A | N/A | N/A | N/A | 6853 | 503.1 | 5881 | 2.5 | 5861 | 3.6 |
| G62 | 7000 | 14000 | 4497 | 49.7 | N/A | N/A | N/A | N/A | 5685 | 242.4 | 4983 | 3.4 | 5086 | 2.7 |
| G63 | 7000 | 41459 | 33861 | 73.4 | N/A | N/A | N/A | N/A | 35322 | 658.5 | 32868 | 4 | 31926 | 1.9 |
| G64 | 7000 | 41459 | 8773 | 73.4 | N/A | N/A | N/A | N/A | 10443 | 186.9 | 8911 | 2.8 | 9171 | 2.5 |
| G65 | 8000 | 16000 | 5212 | 59.6 | N/A | N/A | N/A | N/A | 6490 | 324.7 | 5735 | 3.5 | 5775 | 2.6 |
| G66 | 9000 | 18000 | 5948 | 69.0 | N/A | N/A | N/A | N/A | 7416 | 542.5 | 6501 | 5.4 | 6610 | 3.9 |
| G67 | 10000 | 20000 | 6545 | 79.0 | N/A | N/A | N/A | N/A | 8086 | 756.7 | 7001 | 3.5 | 7259 | 4.1 |
| G70 | 10000 | 9999 | 9718 | 74.8 | N/A | N/A | N/A | N/A | 9999 | 7.8 | 9982 | 4.2 | 9971 | 2.5 |
| G72 | 10000 | 20000 | 6612 | 79.2 | N/A | N/A | N/A | N/A | 8192 | 271.2 | 7210 | 5.1 | 7297 | 3.5 |
| G77 | 14000 | 28000 | 9294 | 142.3 | N/A | N/A | N/A | N/A | 11578 | 154.9 | 10191 | 8.6 | 10329 | 8.5 |
| G81 | 20000 | 40000 | 13098 | 241.1 | N/A | N/A | N/A | N/A | 16321 | 331.2 | 14418 | 20.2 | 14464 | 9.7 |

# D   EVALUATION ON GRAPH COLORING DATASET

To further verify the performance of `ROS`, we conduct numerical experiments on the publicly available COLOR dataset (three benchmark instances: anna, david, and huck). The COLOR dataset provides dense problem instances with relatively large known chromatic numbers ($\chi \sim 10$), which is suitable for testing the performance on Max-$k$-Cut tasks. As reported in Tables 5 and 6, `ROS` achieves superior performances across nearly all settings with the least computational time (in seconds).

Table 5: Objective values returned by each method on the COLOR dataset.

| Methods | anna | | david | | huck | |
|---|---|---|---|---|---|---|
| | $k = 2$ | $k = 3$ | $k = 2$ | $k = 3$ | $k = 2$ | $k = 3$ |
| MD | 339 | **421** | 259 | 329 | 184 | 242 |
| PI-GNN | 322 | - | 218 | - | 170 | - |
| ecord | **351** | - | **267** | - | **191** | - |
| ANYCSP | **351** | - | **267** | - | **191** | - |
| ROS | **351** | 421 | 266 | **338** | **191** | **244** |

Table 6: Computational time for each method on the COLOR dataset.

| Methods | anna | | david | | huck | |
|---|---|---|---|---|---|---|
| | $k = 2$ | $k = 3$ | $k = 2$ | $k = 3$ | $k = 2$ | $k = 3$ |
| MD | 2.75 | 2.08 | 2.78 | 2.79 | 2.62 | 2.82 |
| PI-GNN | 93.40 | - | 86.84 | - | 102.57 | - |
| ecord | 4.87 | - | 4.74 | - | 4.88 | - |
| ANYCSP | 159.35 | - | 138.14 | - | 127.36 | - |
| ROS | **1.21** | **1.23** | **1.18** | **1.15** | **1.11** | **1.10** |

# E   ABLATION STUDY

## E.1   MODEL ABLATION

We conducted additional ablation studies to clarify the contributions of different modules.

**Effect of Neural Networks:** We consider two cases: (i) replace GNNs by multi-layer perceptrons (denoted by `ROS-MLP`) in our ROS framework and (ii) solve the relaxation via mirror descent (denoted by `MD`). Experiments on the Gset dataset show that `ROS` consistently outperforms `ROS-MLP` and `MD`, highlighting the benefits of using GNNs for the relaxation step.

**Effect of Random Sampling:** We compared `ROS` with `PI-GNN`, which employs heuristic rounding instead of our random sampling algorithm. Results indicate that `ROS` generally outperforms `PI-GNN`, demonstrating the importance of the sampling procedure.

These comparisons, detailed in Tables 7 and 8, confirm that both the GNN-based optimization and the random sampling algorithm contribute significantly to the overall performance.

## E.2   SAMPLE EFFECT ABLATION

We investigated the effect of the number of sampling iterations and report the results in Tables 9, 10, 11, and 12.

**Objective Value** (Table 9, Table 11): The objective values stabilize after approximately 5 sampling iterations, demonstrating strong performance without requiring extensive sampling.

**Sampling Time** (Table 10, Table 12): The time spent on sampling remains negligible compared to the total computational time, even with an increased number of samples.

Table 7: Objective values returned by each method on Gset.

| Methods | G70 | | G72 | | G77 | | G81 | |
|---------|-------|-------|-------|-------|-------|-------|-------|-------|
| | $k = 2$ | $k = 3$ | $k = 2$ | $k = 3$ | $k = 2$ | $k = 3$ | $k = 2$ | $k = 3$ |
| ROS-MLP | 8867 | 9943 | 6052 | 6854 | 8287 | 9302 | 12238 | 12298 |
| PI-GNN | 8956 | – | 4544 | – | 6406 | – | 8970 | – |
| MD | 8551 | 9728 | 5638 | 6612 | 7934 | 9294 | 11226 | 13098 |
| ROS | 8916 | 9971 | 6102 | 7297 | 8740 | 10329 | 12332 | 14464 |

Table 8: Computational time for each method on Gset.

| Methods | G70 | | G72 | | G77 | | G81 | |
|---------|-------|-------|--------|-------|--------|--------|--------|--------|
| | $k = 2$ | $k = 3$ | $k = 2$ | $k = 3$ | $k = 2$ | $k = 3$ | $k = 2$ | $k = 3$ |
| ROS-MLP | 3.49 | 3.71 | 3.93 | 4.06 | 8.39 | 9.29 | 11.98 | 16.97 |
| PI-GNN | 34.50 | – | 253.00 | – | 349.40 | – | 557.70 | – |
| MD | 54.30 | 74.80 | 44.20 | 79.20 | 66.00 | 142.30 | 130.80 | 241.10 |
| ROS | 3.40 | 2.50 | 3.90 | 3.50 | 8.10 | 8.50 | 9.30 | 9.70 |

These results highlight the efficiency of our sampling method, achieving stable and robust performance with little computational cost.

## F    COMPARISON AGAINST ADDITIONAL BASELINES ON GSET

We have conducted additional experiments comparing ROS against ANYCSP and ECORD on the Gset benchmark for Max-Cut, focusing on both solution quality and computational efficiency. The results are presented below.

**Results on Gset (unweighted) with Edge Weights of $\pm 1$:** Tables 13 and 14 present the comparison of objective values and inference times for each method on unweighted Gset instances. Although ANYCSP achieves marginally better objective values, its computational time is considerably longer. ECORD, on the other hand, fails to generate competitive solutions. In contrast, our ROS framework strikes an optimal balance, delivering high-quality solutions in a fraction of the time required by ANYCSP.

Weighted Max-Cut has numerous applications, including but not limited to physics (De Simone et al., 1995), power networks (Hojny et al., 2021), and data clustering (Poland & Zeugmann, 2006). To demonstrate the capability of our ROS framework in solving general Max-$k$-Cut problems, we evaluate the performance of ROS, ANYCSP, and ECORD in this context.

**Results on Gset with Arbitrary Edge Weights.** We modified the four largest Gset instances (G70, G72, G77, and G81) to incorporate arbitrary edge weights. Specifically, we perturb the edge weights uniformly within the range $[-10\%, 10\%]$ for each Gset benchmark and generate 10 instances. The averaged results, along with their standard deviations, summarized in Tables 15 and 16, reveal the limitations of both ANYCSP and ECORD in this setting. ANYCSP fails to produce meaningful solutions due to its CSP-based formulation, which overlooks edge weights, and ECORD again demonstrates poor performance. In contrast, ROS consistently generates high-quality solutions while maintaining computational efficiency, showcasing its robustness and versatility across different scenarios.

Table 9: Objective value results corresponding to the times of sample $T$ on Gset.

| $T$ | G70 | | G72 | | G77 | | G81 | |
|---|---|---|---|---|---|---|---|---|
| | $k=2$ | $k=3$ | $k=2$ | $k=3$ | $k=2$ | $k=3$ | $k=2$ | $k=3$ |
| 1 | 8911 | 9968 | 6100 | 7305 | 8736 | 10321 | 12328 | 14460 |
| 5 | 8915 | 9969 | 6102 | 7304 | 8740 | 10326 | 12332 | 14462 |
| 10 | 8915 | 9971 | 6102 | 7305 | 8740 | 10324 | 12332 | 14459 |
| 25 | 8915 | 9971 | 6102 | 7307 | 8740 | 10326 | 12332 | 14460 |
| 50 | 8915 | 9971 | 6102 | 7307 | 8740 | 10327 | 12332 | 14461 |
| 100 | 8916 | 9971 | 6102 | 7308 | 8740 | 10327 | 12332 | 14462 |

Table 10: Sampling time results corresponding to the times of sample $T$ on Gset.

| $T$ | G70 | | G72 | | G77 | | G81 | |
|---|---|---|---|---|---|---|---|---|
| | $k=2$ | $k=3$ | $k=2$ | $k=3$ | $k=2$ | $k=3$ | $k=2$ | $k=3$ |
| 1 | 0.0011 | 0.0006 | 0.0011 | 0.0006 | 0.0020 | 0.0010 | 0.0039 | 0.0020 |
| 5 | 0.0030 | 0.0029 | 0.0029 | 0.0030 | 0.0053 | 0.0053 | 0.0099 | 0.0098 |
| 10 | 0.0058 | 0.0059 | 0.0058 | 0.0058 | 0.0104 | 0.0104 | 0.0196 | 0.0196 |
| 25 | 0.0144 | 0.0145 | 0.0145 | 0.0145 | 0.0259 | 0.0260 | 0.0489 | 0.0489 |
| 50 | 0.0289 | 0.0289 | 0.0288 | 0.0289 | 0.0517 | 0.0518 | 0.0975 | 0.0977 |
| 100 | 0.0577 | 0.0577 | 0.0576 | 0.0578 | 0.1033 | 0.1037 | 0.1949 | 0.1953 |

Table 11: Objective value results corresponding to the times of sample $T$ on random regular graphs.

| $T$ | $n=100$ | | $n=1000$ | | $n=10000$ | |
|---|---|---|---|---|---|---|
| | $k=2$ | $k=3$ | $k=2$ | $k=3$ | $k=2$ | $k=3$ |
| 1 | 127 | 245 | 1293 | 2408 | 12856 | 24103 |
| 5 | 127 | 245 | 1293 | 2410 | 12863 | 24103 |
| 10 | 127 | 245 | 1293 | 2410 | 12862 | 24103 |
| 25 | 127 | 245 | 1293 | 2410 | 12864 | 24103 |
| 50 | 127 | 245 | 1293 | 2410 | 12864 | 24103 |
| 100 | 127 | 245 | 1293 | 2410 | 12864 | 24103 |

Table 12: Sampling time results corresponding to the times of sample $T$ on random regular graphs.

| $T$ | $n=100$ | | $n=1000$ | | $n=10000$ | |
|---|---|---|---|---|---|---|
| | $k=2$ | $k=3$ | $k=2$ | $k=3$ | $k=2$ | $k=3$ |
| 1 | 0.0001 | 0.0001 | 0.0001 | 0.0001 | 0.0006 | 0.0006 |
| 5 | 0.0006 | 0.0006 | 0.0007 | 0.0007 | 0.0030 | 0.0030 |
| 10 | 0.0011 | 0.0011 | 0.0014 | 0.0013 | 0.0059 | 0.0059 |
| 25 | 0.0026 | 0.0026 | 0.0033 | 0.0031 | 0.0145 | 0.0145 |
| 50 | 0.0052 | 0.0052 | 0.0065 | 0.0060 | 0.0289 | 0.0289 |
| 100 | 0.0103 | 0.0103 | 0.0128 | 0.0122 | 0.0577 | 0.0578 |

Table 13: Objective values returned by each method on Gset for $k=2$.

| Methods | G70 | G72 | G77 | G81 |
|---|---|---|---|---|
| Ecord | 5137 | 206 | 382 | 358 |
| ANYCSP | **9417** | **6826** | **9694** | **13684** |
| ROS | 8916 | 6102 | 8740 | 12332 |

Table 14: Computational time for each method on Gset for $k = 2$.

| Methods | G70 | G72 | G77 | G81 |
|---|---|---|---|---|
| Ecord | **1.4** | **1.2** | **1.7** | **2.4** |
| ANYCSP | 180.0 | 180.0 | 180.0 | 180.0 |
| ROS | 3.4 | 3.9 | 8.1 | 9.3 |

Table 15: Objective value on Gset with arbitrary edge weights for $k = 2$.

| Methods | G70 | G72 | G77 | G81 |
|---|---|---|---|---|
| Ecord | $5154.28 \pm 28.26$ | $254.46 \pm 37.22$ | $344.79 \pm 29.73$ | $280.09 \pm 33.11$ |
| ANYCSP | $5198.87 \pm 69.76$ | $-15.57 \pm 57.88$ | $81.76 \pm 69.97$ | $33.49 \pm 50.31$ |
| ROS | $\mathbf{8941.80 \pm 17.79}$ | $\mathbf{6165.62 \pm 50.81}$ | $\mathbf{8737.59 \pm 114.24}$ | $\mathbf{12325.85 \pm 87.98}$ |

Table 16: Computational time comparison on Gset with arbitrary edge weights for $k = 2$.

| Methods | G70 | G72 | G77 | G81 |
|---|---|---|---|---|
| Ecord | $3.39 \pm 0.11$ | $\mathbf{3.43 \pm 0.12}$ | $\mathbf{3.89 \pm 0.03}$ | $\mathbf{4.76 \pm 0.05}$ |
| ANYCSP | $180.56 \pm 0.08$ | $180.52 \pm 0.03$ | $180.57 \pm 0.11$ | $180.68 \pm 0.10$ |
| ROS | $\mathbf{2.97 \pm 0.56}$ | $4.32 \pm 2.03$ | $6.97 \pm 3.58$ | $9.24 \pm 3.43$ |

