# OpenReview forum: "ROS: A GNN-based Relax-Optimize-and-Sample Framework for Max-$k$-Cut Problems"
_ICLR.cc/2025/Conference — Submitted to ICLR 2025_

### Official Review · Reviewer_wn14 · 2024-11-03

**Soundness:** 1
**Presentation:** 2
**Contribution:** 2
**Rating:** 5
**Confidence:** 4

**Summary:**

The paper proposes a continuous relaxation of the max-$k$-cut problem that uses a categorical variable for each node in the graph to assign it to one of $k$ categories. This relaxation is then used as the loss function to train a graph neural network to solve the problem. The paper proposes an additional pretraining step that can help improve the model's performance. To obtain discrete solutions from the relaxation the paper proposes a straightforward sampling of the categorical variables.

Since the relaxation can be viewed as an expectation over the categorical variables, the globally optimal solutions are preserved which justifies the use of this particular relaxation. The paper shows competitive experimental results for max-k-cut with different k values on benchmark data.

**Strengths:**

- The choice of relaxation for the problem is quite sensible.
- Experimentally, the method works fairly well.
- The paper studies the effects of pre-training on the generalization of the model.

**Weaknesses:**

- The discussion of related work is inadequate. For instance, there have been several works in neural combinatorial optimization that focus on max-cut which are not discussed (such as [1,2] or more recently [3]).  The unsupervised approach via a relaxation resembles the approach in [4] and other works in that spirit. Those need to be brought up and discussed and the contribution of this work needs to be explained in the context of this existing literature.

- In my view, the main paper needs to contain a succint presentation of the Gset experiments. As it is, just looking at the results on a few graphs is certainly not convincing.

- This connects to my comment about related work, but the experimental comparisons are lacking. Several important baselines are missing (e.g., [1,2]) for the max-cut problem. Showing how the proposed method performs against strong baselines like that is essential for a paper that focuses on 1 problem.

- The contribution in the paper is somewhat limited, since similar approaches have been proposed in the literature for several other problems. It's hard to point out something that stands out, with the exception of maybe the empirical results in some cases for larger values of $k$.

- Model ablations (e.g., comparison with a simple MLP that uses the same relaxation) would help establish the usefulness of this specific architecture.

- The pertaining approach to the problem seems to have mixed results and is not that convincing.

Overall, I don't think this is a bad paper, but I  don't think the contribution is strong enough to warrant acceptance. I start with a tentative score and I am willing to reconsider after the rebuttal.

1. Barrett, Thomas D., Christopher WF Parsonson, and Alexandre Laterre. "Learning to solve combinatorial graph partitioning problems via efficient exploration." arXiv preprint arXiv:2205.14105 (2022).
2. Tönshoff, Jan, et al. "One model, any csp: Graph neural networks as fast global search heuristics for constraint satisfaction." arXiv preprint arXiv:2208.10227 (2022).
3. Nath, Ankur, and Alan Kuhnle. "A Benchmark for Maximum Cut: Towards Standardization of the Evaluation of Learned Heuristics for Combinatorial Optimization." arXiv preprint arXiv:2406.11897 (2024).
4. Karalias, Nikolaos, and Andreas Loukas. "Erdos goes neural: an unsupervised learning framework for combinatorial optimization on graphs." Advances in Neural Information Processing Systems 33 (2020): 6659-6672.

**Questions:**

- Since the value computed is an expectation, couldn't the method of conditional expectation be used to decode a discrete solution from the continuous one? (like in ref 4.)

---

### Official Review · Reviewer_BWJY · 2024-11-04

**Soundness:** 2
**Presentation:** 2
**Contribution:** 2
**Rating:** 6
**Confidence:** 2

**Summary:**

This paper proposes a neural-network based framework to solve the max-k-cut problem. Specifically, the authors try to first solve the relaxed problem and then generate integer solutions via random sampling.

**Strengths:**

The authors ask a very good research question --- can we tackle the NP-hard problem, more specifically the max-k-cut problem, by using a black-box neural network?

**Weaknesses:**

1. I am not totally convinced whether it is the neural network or the random sampling part that is driving the final performance. More careful ablation and perturbation studies are needed in order to shed light on different choices made by the authors. Right now, the whole method looks like a black-box to me. It somehow works, but I have no idea what effect each procedure has for the final performance.

**Questions:**

1. How much of an effect does random sampling play? Can you also report results (for Table 1 and Table 2) when you only do random samplings for 1 time, 5 times, 10 times, 25 times, and 50 times?

2. The mirror descent (MD) method is only solving the relaxed problem? How does MD give you the final integer solution? Are you also doing random sampling on this method? If so, how many times of random sampling are you doing?

3. Can you share an anonymous link of your code to the AC so that AC can check the reproducibility of this work?

---

### Official Review · Reviewer_Vxrh · 2024-11-04

**Soundness:** 3
**Presentation:** 3
**Contribution:** 2
**Rating:** 6
**Confidence:** 2

**Summary:**

The goal of the paper is to map each discrete variable, which can take k values, into a vector of 𝑘 numbers that lie between 0 and 1, such that their sum equals 1 (a probability vector), and each vector slot represents a mutually exclusive choice. Then, a graph neural network is used to find these probability simplices that optimize the objective function. Each variable is then assigned back to a discrete value according to the probability distribution, independently of other variables. The authors demonstrate the application of this method to the Maximum-k-Cut problem.

**Strengths:**

1. The framework can be easily extended to other combinatorial problems like Graph Coloring, Maximum Cover.
2. The writing and clarity is good.
3. This approach is scalable to large instances.

**Weaknesses:**

1. Since the problem can be extended to other graph-related combinatorial problems where vertices need to be separated into groups, the author should consider including additional problems as well.
2. I do not understand what Theorem 1 contributes. It states that there should be a globally optimal integer solution if we find the globally optimal solution to the continuous problem. However, the GNN does not guarantee an optimal solution for the continuous problem.
3. The idea of generating probability distributions for variables (soft assignment) followed by hard assignments (random sampling) is not new. See the papers: 1) Erdos Goes Neural: An Unsupervised Learning Framework for Combinatorial Optimization on Graphs and 2) Graph Neural Networks for Maximum Constraint Satisfaction.
4. While Gset is a well-known benchmark for Maximum Cut (k=2), its difficuility for k>2 is unknown. Besides the author only evaluated on unweighted instances of Gset. Additionally, the author only evaluated on unweighted instances of Gset. I suggest the author consider hard instances of Graph Coloring, which can easily be mapped to the Maximum-k-Cut problem.
5. Given the performance of the neural baseline PI-GNN on Maximum Cut, it is not a strong baseline for this problem. There are other algorithms that perform better on Gset, such as One Model, Any CSP: Graph Neural Networks as Fast Global Search Heuristics for Constraint Satisfaction.
6. The figures can be improved by increasing the font size.

**Questions:**

1. What does it mean by the consistency of function values between the continuous solution and its sampled discrete counterpart (line 102)?
2. In the definition of Max-k-Cut and experiments, you only consider non-negative weights ? Any particular reason for that ?
3. What is the value of T in these experiments? (line 240)

---

### Meta-Review · Area_Chair_EmKj · 2024-12-18

**Metareview:**

This paper tries to address the Max-k-Cut problem, whose existing solutions often require relaxation techniques that lack solution guarantees. To improve this, the Relax-Optimize-and-Sample (ROS) framework is proposed, combining relaxation, graph neural networks, and a sampling-based algorithm to efficiently solve the problem.  Consistency between the continuous and rounded solutions is established by analyzing the geometric landscape. Extensive experiments show that ROS outperforms state-of-the-art algorithms, scaling effectively to large graphs and demonstrating strong generalization capabilities for both in-distribution and out-of-distribution instances.

The paper addresses an important problem, and the solution appears interesting and effective.  The overall relaxation approach uses pretty standard techniques, with some interesting theoretical results.  Then the use of GNN sets the evaluation back to empirical.  My major concern is the lack of comparison with ECORD and ANYCSP for $k \ge 3$.  Although the rebuttal provided some insights, it will be important to see the comparison results before the paper can be published.

**Additional Comments On Reviewer Discussion:**

The rebuttal has been noted by the reviewers and have been taken into account by the AC in the recommendation of acceptance/rejection.

---

### Decision · Program_Chairs · 2025-01-22

Reject